# Frontal cortex activity during the production of diverse social communication calls in marmoset monkeys

Lingyun Zhao [1,2] ✉ & Xiaoqin Wang [1] ✉

Vocal communication is essential for social behaviors in humans and non-human primates. While the frontal cortex is crucial to human speech production, its role in vocal production in non-human primates has long been questioned. It is unclear whether activities in the frontal cortex represent diverse vocal signals used in non-human primate communication. Here we studied single neuron activities and local field potentials (LFP) in the frontal cortex of male marmoset monkeys while the animal engaged in vocal exchanges with conspecifics in a social environment. We found that both single neuron activities and LFP were modulated by the production of each of the four major call types. Moreover, neural activities showed distinct patterns for different call types and theta-band LFP oscillations showed phase-locking to the phrases of twitter calls, suggesting a neural representation of vocalization features. Our results suggest important functions of the marmoset frontal cortex in supporting the production of diverse vocalizations in communication.

Many non-human primates (NHPs) use species-specific vocalizations during social communication. Field studies and laboratory experiments have shown that species-specific vocalizations provide important functions for their social behaviors[1]. Evidence has been found that the production and use of species-specific vocalizations resemble some of the rudimentary features in human speech[1–3]. However, the neural circuits responsible for vocal production in NHPs have remained elusive. Studies in humans have shown that the production of emotional vocalizations (e.g., cry and laughter) is primarily controled by the structures on the medial side of the brain, including the anterior cingulate cortex and periaqueductal gray (PAG)[4]. In contrast, the production of communicative signals (i.e., speech) is primarily controlled by the structures on the lateral side of the brain, including Broca's area and multiple premotor and motor regions in the frontal cortex[4]. These lateral structures are especially important for the timing[5,6] and the generation of distinct acoustic structures[7–10] in human speech. A large body of previous studies in NHPs focused on vocalizations that are induced by operant conditioning or electrical stimulation[11,12]. The brain structures identified in these studies are largely on the medial side, such as the anterior cingulate cortex and PAG, similar to what is responsible for emotional vocalizations in humans. The prefrontal cortex and the premotor cortex are found to be active by conditioned vocalizations but not by spontaneous or self-initiated vocal production[13,14]. It has been a long-standing question whether these lateral structures in the frontal cortex of NHPs are involved in natural vocal production and whether they exhibit any call-specific activation.

There has been growing evidence to suggest that the common marmoset (*Callithrix jacchus*), a New World monkey species, may have a higher level of vocal plasticity and flexibility than many other NHPs[15]. For example, it has been shown that the vocal development in juvenile marmosets was influenced by social feedback from parents through vocal interactions[16–20]. Adult marmosets demonstrated flexibility in and voluntary control of some aspects of vocal production, including the initiation time[2], acoustic structures[21–23] and

[1]Laboratory of Auditory Neurophysiology, Department of Biomedical Engineering, The Johns Hopkins University School of Medicine, Baltimore, MD 21205, USA. [2]Present address: Department of Neurological Surgery, University of California, San Francisco, CA 94158, USA. ✉e-mail: lingyun.zhao@ucsf.edu; xiaoqin.wang@jhu.edu

vocal turn-taking[3,24]. Acoustic structures of marmoset vocalizations were also found to be modulated by social contexts[25–27].

Using an antiphonal calling paradigm[28,29], recent studies have found evidence that neural activities in premotor and prefrontal cortices of the marmoset were modulated by phee calls, a call type produced mostly in isolation[30–33]. Marmosets typically produce more than 15 types of vocalizations during natural social communication[34–36]. It is not yet clear whether the frontal cortex encodes call-specific signals in the generation of these social communication calls. A recent anatomical study showed strong descending projections from the marmoset premotor cortex to downstream vocal control structures, suggesting potential functions of the premotor cortex in vocal motor skills[37]. Given the rich vocal behaviors exhibited by marmosets, we hypothesized that the frontal cortex of marmosets is involved in the production of all call types especially the calls used in natural vocal communication. If this were true, one would like to know what specific functional roles that the frontal cortex plays in the generation of these calls. For example, the frontal cortex may produce a high-level neural signal to instruct subcortical structures to create the vocal structure of particular call types, in which case the activity of the frontal cortex may not represent the vocal structure of different types of calls. Alternatively, the frontal cortex may produce specific vocal production signals that are associated with individual call types' vocal structure that are sent to the subcortical structures for execution. A challenge in studying these questions is the ability to record from freely moving marmosets in a social environment in which (and only in which) the variety of vocal communication calls are produced.

In the present study, we performed our experiments in a marmoset breeding colony room, in which dozens of marmosets were housed in individual or group cages. Marmosets in the colony generated all types of calls within their vocal repertoire and made frequent vocal exchanges with other marmosets in the room. To enable neural recording experiments in such an environment, we developed techniques to overcome challenges from noise interference. We focused on four major types of calls used in marmoset communication (twitter, trill, trillphee, phee), each of which has distinct acoustic structures[34]. Our results show that local field potentials (LFPs) and single neuron activities are both differentially modulated by call types. Further analysis shows that the neural activities distinguish the call types being produced and reflect temporal dynamics of the vocal structures within a call. These findings have important implications on the functional role of the marmoset frontal cortex in vocal production and communications.

## Results

We recorded single neuron activities and LFP from the left frontal cortex of two marmosets while they were freely roaming in a housing cage in a marmoset colony room and engaged in vocal exchanges with conspecifics housed in the same room (Fig. 1, see Methods). One of the subjects was implanted with a 32-channel electrode array (Subject M93A, brown rectangle in Fig. 1c) and the other subject was implanted with a 16-channel electrode array (Subject M9606, cyan square in Fig. 1c). In each recording session, a wireless headstage was attached to the chronically implanted electrode array and the experimental subject engaged in natural behaviors, including eating, drinking, playing and vocalizing without being interfered with by the recording setup. There was no other marmoset housed in the same cage with the experimental subject during the session, and the experimental subject was able to see and hear other marmosets in the neighboring cages and other locations of the room, including the subject's family members. The experimental subject produced all types of vocalizations within the vocal repertoire of marmosets and the call types included in analyses were consistent between the experimental subjects and the general marmoset population[34]. While the colony room provided a much richer social environment for the marmosets, it also possessed

challenges for acoustic and neural recordings. To ensure reliable recordings, we developed a new apparatus for wireless neural recording (Fig. 1a, b)[38]. We also developed a parabolic microphone system to capture vocalizations produced by the experimental subject from the noisy background (Fig. 1d–f, see Methods). This was crucial because some types of marmoset vocalizations like trills could be easily masked by the background noises if ordinary microphones were used (Fig. 1f). Neural signals and vocalizations were continuously recorded during an experimental session for two to five hours each day.

### Beta-band suppression related to vocal production

We analyzed LFP signals recorded while marmosets produced vocalizations during social communication. LFP activity in the beta-band (12–30 Hz) has long been observed in the frontal motor areas during resting and motor movements in humans and animals[39,40]. Beta-band suppression was found during motor preparation and execution of voluntary movements and was thought to be related to the synchronization of neural activities[39–42]. We first examined whether there was any modulation in the motor areas of the marmoset frontal cortex comparable to what was often seen for voluntary movement (Fig. 2).

We focused on the four major types of vocalizations that were most frequently produced by marmosets in captivity with distinct spectrotemporal features (Fig. 2a–d). Three of the four call types are narrowband (phee, trill and trillphee) whereas the other one is wideband (twitter). A phee call is a loud tone-like long call with slowly changing linear frequency modulation and usually composed of one or more phrases (Fig. 2a). A trill call is a relatively short call with low intensity and characterized by sinusoidal frequency modulation (Fig. 2b). A twitter call is composed of multiple short phrases of sharply rising frequency modulations (Fig. 2c). A trillphee call starts with sinusoidal frequency modulation like in a trill call and then changes into linear frequency modulation like a phee call (Fig. 2d).

Figure 2e top panel shows the time-frequency representation of LFP associated with phee calls recorded from one example site in the motor cortex (indicated by a blue circle on Fig. 2i). The shaded bar near the top indicates the average duration of recorded phees (truncated at 2.5 sec). The strongest modulation was observed in the beta band (Fig. 2e, upper panel). To quantify this modulation, we compared the averaged power of the beta-band LFP (12–30 Hz) relative to the baseline window ([−3,−1] sec) (Fig. 2e, bottom). Beta-band LFP showed a decrease in power before and during the production of phee calls (vocal onset at 0 sec, Fig. 2e), consistent with the beta-band suppression in motor areas reported in the previous literature. The suppression started before the vocal onset and reached the maximum near the vocal onset. The earliest time at which the beta-band power showed significant suppression for this site is at −0.485 sec (Fig. 2e bottom panel, indicated by a downward black triangle, $p < 0.05$, two-sided signed-rank test). Similar beta-band suppression was observed in trill, twitter and trillphee calls at various recording sites (Fig. 2f–h). In all cases, the beta-band power started to decrease prior the vocal onset (−0.045 sec for trills, −0.055 sec for twitters, −0.010 sec for trillphees, $p < 0.05$, two-sided signed-rank test) and reached the largest suppression shortly after the vocal onset (Fig. 2f–h).

To compare the vocalization-related beta-band suppression across cortical regions, we calculated the beta-band LFP power for each site within an analysis window near the vocal onset ([−0.1, 0.2] sec, Fig. 2e–h, gray bars below the bottom panels). The magnitude and significance of the suppression effect for all four call types across recording sites are illustrated on Fig. 2i–p for the two experimental subjects (sites with significant suppression indicated by black circles, $p < 0.05$, two-sided signed-rank test). For phee, trill and twitter calls, the sites with significant beta-band suppression scattered across multiple regions of the frontal cortex (Fig. 2i–k, m–o), whereas the

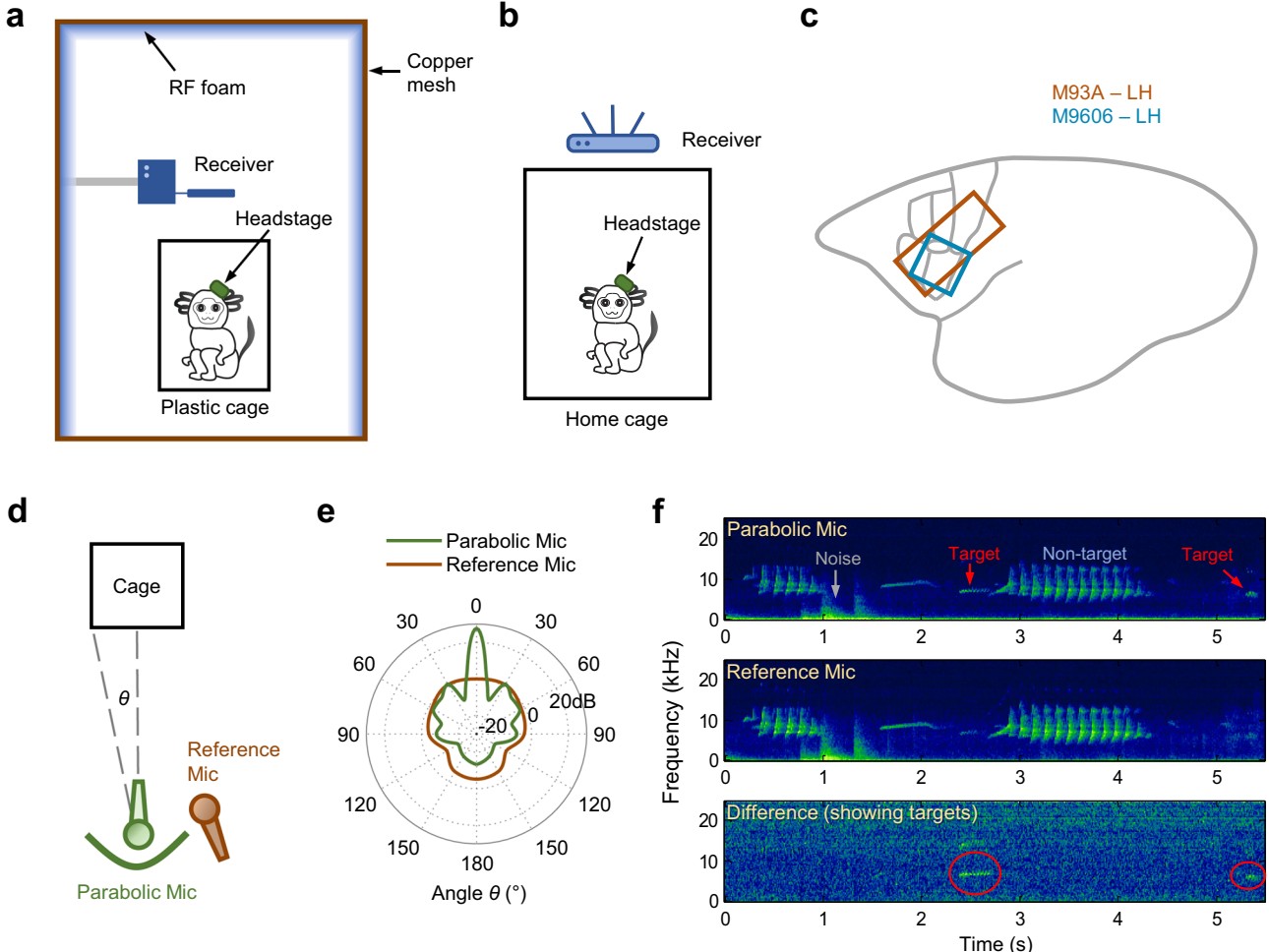

**Fig. 1 | Experimental setup in the marmoset colony. a** Schematic of the wireless neural recording setup (side view) using the analog system (for Subject M9606). The subject and the wireless receiver were enclosed by a shielded booth to minimize interference during wireless signal transmission. The shielded booth was constructed with copper mesh walls and lined with radio frequency (RF) absorption foam. **b** Schematic of the wireless neural recording setup (side view) using the digital system (for Subject M93A). **c** Illustration of the areas (colored rectangles) covered by electrode arrays within the marmoset cortex for the two subjects. M93A: 32 channels. M9606: 16 channels. LH: left hemisphere. The gray lines near the rectangles outline the borders of several Brodmann areas covered by the arrays (see Fig. 2i–p). **d** Schematic of the directed acoustic recording setup (top view). Microphones were placed in front of the cage where the subject (target) was housed. The green curve illustrates the parabolic reflector. **e** Pick-up pattern (gain at different angles) of the two types of microphones at 8 kHz. The gain of the parabolic mic peaks at the zero-degree angle (front-facing direction) and quickly drops as the angle increases (side directions). Within a certain angle range (a narrow range near front-facing direction), the gain of the parabolic mic is higher than the reference mic. **f** An example recording clip (spectrograms) from the parabolic-reference mic pair in the colony. Top: channel from the parabolic mic. Background noise, non-target vocalizations, and the target vocalizations from the subject were recorded. The target vocalization has an enhanced intensity. Middle: channel from the reference mic. Background noise, non-target vocalizations, and weaker target vocalizations were recorded. The intensity of background noise and non-target vocalizations in the parabolic channel is similar to that in the reference channel, whereas the intensity of the target vocalizations is much stronger in the parabolic channel. Bottom: the difference between the parabolic channel and the reference channel. Background noise and non-target vocalizations cancel out. Only target vocalizations remain. Source data are provided as a Source Data file.

sites with significant beta-band suppression appeared to be located in the more caudal regions for trillphee calls in both subjects (Fig. 2l, p).

We then investigated whether the modulation of the beta-band LFP depends on call types or simply indicates the generation of any vocalizations. We compared the beta-band suppression between the four different call types at each recording site. As shown in Fig. 3a by an example recording site, the beta-band modulation shows different temporal profiles for four call types (Fig. 3a, left) and the median beta-band power within the analysis window showed significant differences between call types (Fig. 3a, right, $p < 0.05$ for each pair, Kruskal–Wallis test, *post hoc* analysis with Bonferroni corrections). Phee calls induced the largest suppression and trill induced the smallest. Trillphee has acoustic structures bearing features of both trill and phee and the size of its associated suppression is between that of phee and trill. Figure 3b shows another example recording site with different amplitude and

temporal profiles of beta-band modulation for the four call types. In total, 28 out of 32 sites in Subject M93A and 8 out of 16 sites in Subject M9606 showed different beta-band modulation for different call types.

To quantify the beta-band suppression between different call types, we analyzed the beta-band suppression start time of the sites that showed significant suppression to each call type either before or during the vocalizations (see Methods, with additional analysis windows included). The median suppression start time is before the vocal onset for phee, trill and twitter calls and around the vocal onset for trillphee calls (Fig. 3c, $p < 0.05$, two-sided signed-rank test). Interestingly, the beta-band suppression for phee calls started significantly earlier than the other three call types (Fig. 3c, $p < 0.05$ Kruskal–Wallis test, post hoc analysis with Bonferroni corrections). We further quantify the differences in the temporal profiles of the beta-band

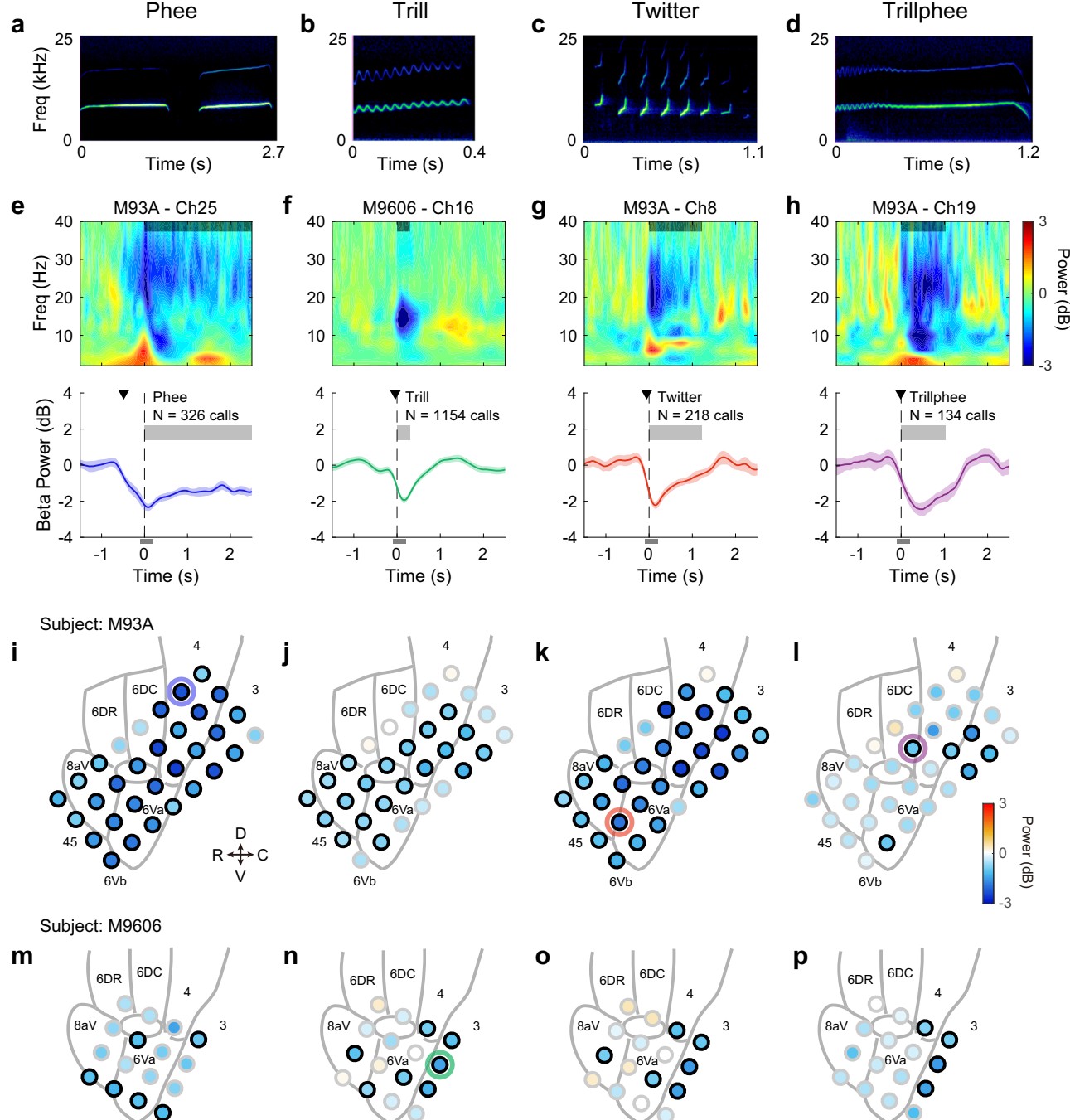

**Fig. 2 | Suppression in the beta-band local field potential (LFP) for four types of marmoset calls. a** Spectrogram of an example phee call (from Subject M93A, with two phrases). **b**–**d** Same format as **a**, spectrograms of trill, twitter and trillphee calls. **e** LFP from one example recording site showing suppression in beta-band for phee call production. The location of the site is indicated by a blue circle in **i**. Top: time-frequency representation of LFP. The shaded horizontal bar illustrates the average call duration (truncated at the axis limit). LFP signals were aligned to the vocal onset (time zero) for each call and the time-frequency representations were averaged across calls. Bottom: LFP power in the beta band (mean ± SEM) relative to that in the baseline window before vocal onset. Vertical dashed line: time zero. Thick gray bar: average call duration. *N* indicates the total number of calls included. The black triangle indicates the suppression start time, i.e., the earliest time at which LFP power showed significant suppression (see Fig. 3c). The thin gray bar below the axis

indicates the analysis window near vocal onset used to quantify the modulation of beta-band power in **i**–**p**. **f**–**h** Same format as **e**, for the three other call types. **i** Spatial distribution of beta-band modulation across all recording sites for phee calls in Subject M93A. Each circle indicates a recording site. The color in the circle indicates the LFP power in the analysis window shown in **e** relative to that in the baseline window before vocal onset. A black border on a circle indicates significant modulation. The blue ring indicates the location of the example site in **e**. Gray lines in the background outline the border of Brodmann areas according to the marmoset brain atlas with their names labeled. A small area near the center without a label is area 8 C. R rostral, C caudal, D dorsal, V ventral. **j**–**l** Same format as **i**, for the other three call types in Subject M93A. **m**–**p** Same format as **i**, for Subject M9606. Source data are provided as a Source Data file.

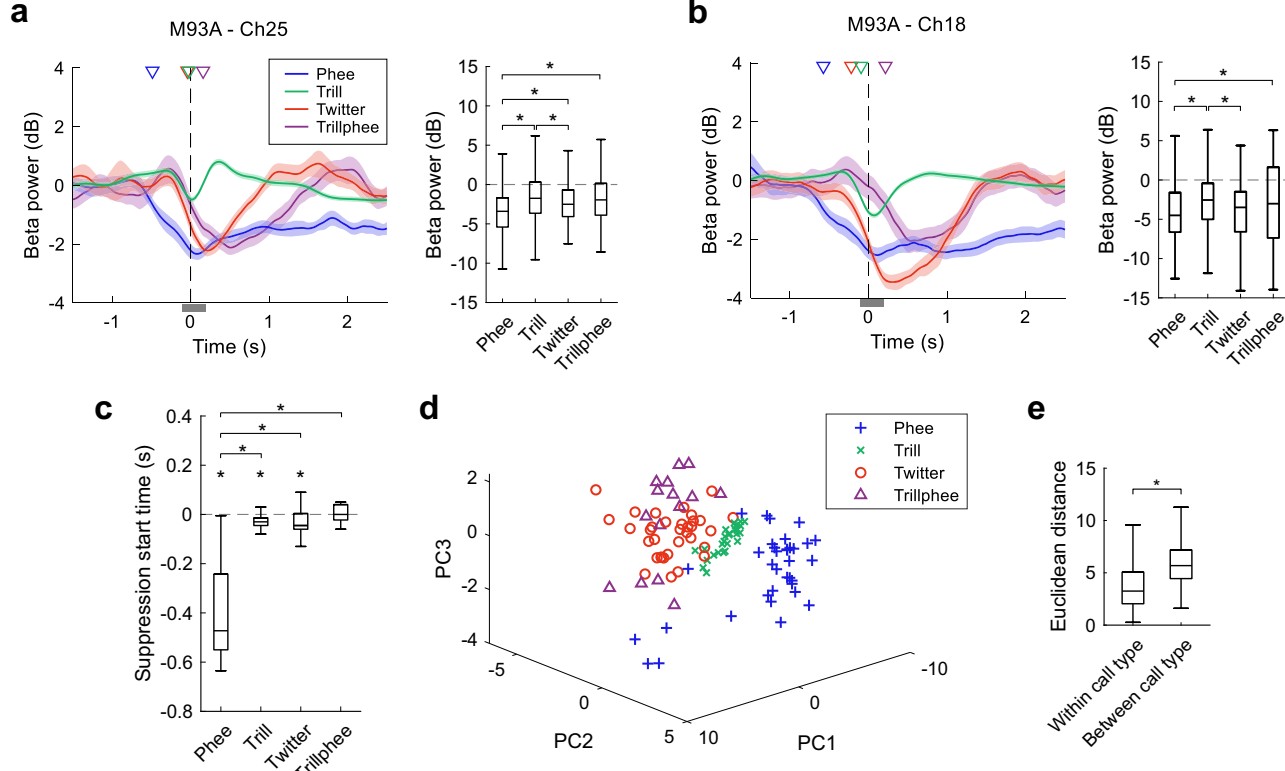

**Fig. 3 | Comparison of the beta-band suppression between four call types.**
**a** Left: LFP power in the beta band (mean ± SEM) from one example recording site for four call types (n = 326 phee calls, n = 2791 trill calls, n = 218 twitter calls, n = 134 trillphee calls). The LFP power is relative to that in the baseline window. The colored triangles indicate the suppression start time, i.e., the earliest time at which LFP power showed significant suppression. The thin gray bar indicates the analysis window to quantify LFP power (see the right panel). Right: LFP power in the beta band (Tukey boxplot) in the analysis window shown in the left panel. Medians: horizontal lines inside the boxes. First and third quartiles: lower and upper borders of the boxes. Inner fences: whiskers outside of the boxes. Outliers are not plotted if any. An asterisk indicates a significant difference between two call types ($\chi^2$ = 99.4, p = 2.1 × 10$^{-21}$, Kruskal–Wallis test, post hoc analysis with Bonferroni corrections: p = 3.6 × 10$^{-21}$, 0.0012, 3.2 × 10$^{-5}$, 0.0042 for phee-trill, phee-twitter, phee-trillphee, and trill-twitter comparisons). **b** Same format as **a**, for another recording site (with the same number of calls as **a**, $\chi^2$ = 63.3, p = 1.2 × 10$^{-13}$, Kruskal–Wallis test, post hoc analysis with Bonferroni corrections: p = 9.7 × 10$^{-12}$, 0.0068, 1.3 × 10$^{-4}$ for phee-trill, phee-trillphee, and trill-twitter comparisons). Boxplot definition is the same as **a**.

**c** Comparison of the suppression start time between call types. Data are from all recording sites showing significant suppression (n = 36 sites for phee, n = 29 sites for trill, n = 39 sites for twitter, n = 15 sites for trillphee). Boxplot definition is the same as **a**. An asterisk above a whisker indicates that the median is significantly lower than zero (p = 1.7 × 10$^{-7}$, 2.2 × 10$^{-4}$, 0.011 for phee, trill, and twitter, two-sided signed-rank test). An asterisk above a bracket indicates a significant difference between two call types ($\chi^2$ = 58.9, p = 1.0×10$^{-12}$, Kruskal–Wallis test, post hoc analysis with Bonferroni corrections: p = 6.5 × 10$^{-8}$, 2.7 × 10$^{-7}$, 5.0 × 10$^{-10}$ for phee-trill, phee-twitter, and phee-trillphee comparisons). **d** Scatter plot of projected LFP traces for each call type using the first three principal components (PCs). Each data point is from a recording site showing significant suppression for the same call type.
**e** Euclidean distance between the projected data points in **d** within (n = 1612 distance samples) or between (n = 4493 distance samples) call types. Boxplot definition is the same as **a**. Asterisk indicates a significant difference (z = −31.4, p = 6.5 × 10$^{-217}$, two-sided rank-sum test). Source data are provided as a Source Data file.

power for the four call types at all recording sites with significant suppression by projecting the temporal profiles onto the principal component (PC) space. The projected samples for each call type are largely separate from each other in the PC space (Fig. 3d). The Euclidean distance between the samples across different call types is larger than that within the same type (Fig. 3e). This suggests that each call type has a characteristic beta-band suppression by which call types can be distinguished at these recording sites. We observed a similar separation of call types using simple features of the temporal profiles, such as the magnitude and time at the peak suppression (Supplementary Fig. 1). It is interesting that the magnitude of the beta-band suppression induced by trillphees spans in a similar range as that induced by phee calls, but the time at the maximum suppression is much later for trillphee than for phees (Supplementary Fig. 1). A possible explanation for this may come from the fact that a trillphee begins with a trill-like first component followed by a phee-like second component. If the beta-band modulation due to the first component follows that for trills, it will induce a small suppression near the vocal onset time. If the modulation due to the second component follows

that for phees, the suppression magnitude will increase in the middle of a trillphee call, thus at a later time after the vocal onset. With the above observations, our data suggest that the beta-band suppression not only reflects the vocal production during social communication but also contains call type specific information both at an individual site and across the sampled cortical regions.

## Theta-band activation and phase lock with call phrases
Besides beta-band suppression, we observed modulations in other frequency bands of LFP as well. Interestingly, a subset of sites showed activation in the theta-band (4–8 Hz). Figure 4a shows an example recording sites showing an increase in LFP power in the theta-band for the production of phee calls. For this site, theta-band activation started before the vocal onset and peaked before it too (Fig. 4a, bottom panel). Theta-band activation is also observed for other call types (Fig. 4b–d). For twitter calls, the theta-band activation lasted for the duration of the calls (Fig. 4c). Overall, the theta-band activation was not as widespread as the beta-band suppression across recording sites, with the majority found for twitter calls (Fig. 4e). When averaging the theta-band power

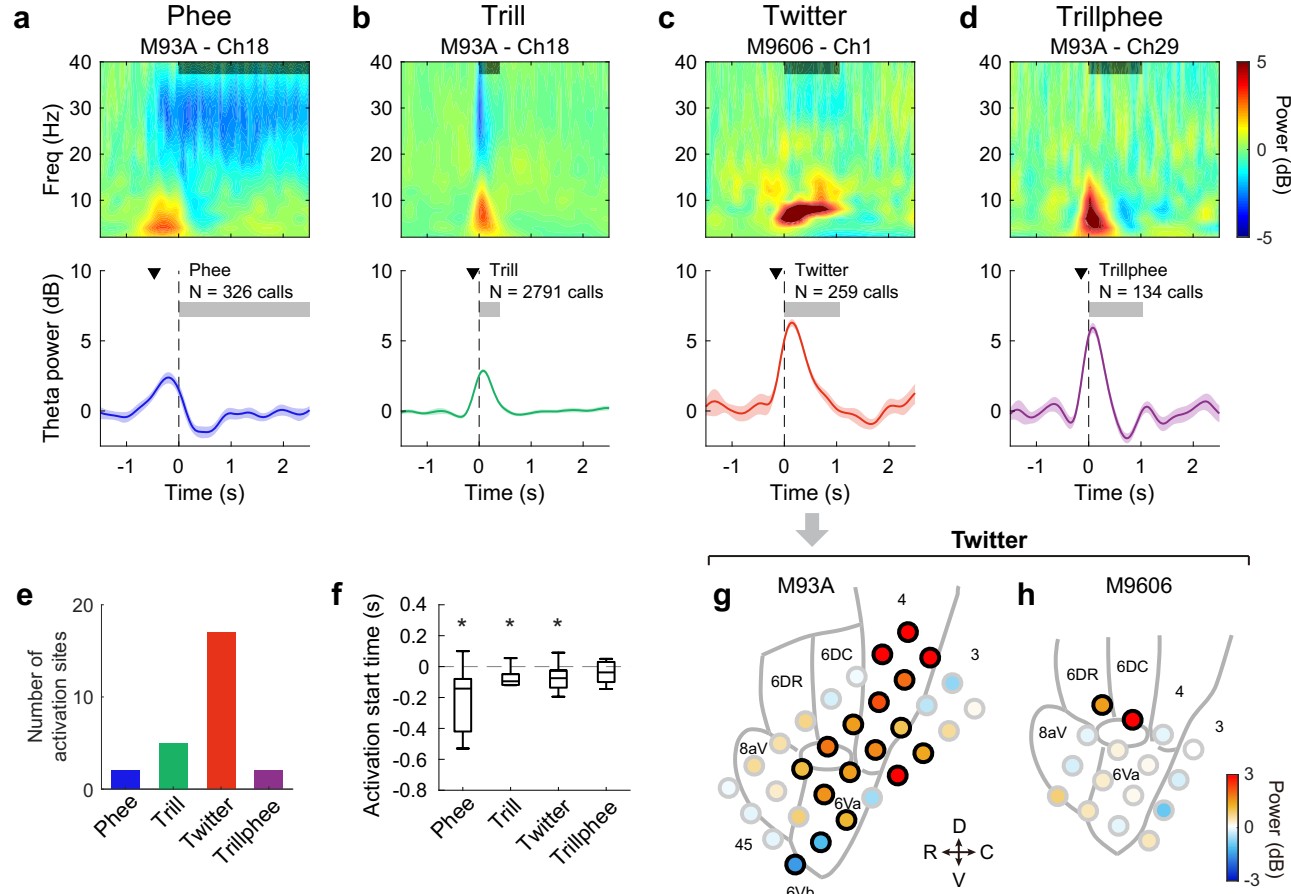

**Fig. 4 | Activation in the theta-band LFP. a** Same format as Fig. 2e, LFP from one example recording site (in Subject M93A) showing activation in theta band when the subject vocalized phee calls. Data shown in the bottom panel reflect mean ± SEM. The black triangle in the bottom panel indicates the activation start time, i.e., the earliest time at which LFP power showed significant activation (see **f**). **b**−**d** Same format as **a**, for trill, twitter, and trillphee calls. **e** Number of sites with theta-band activation within the call duration across both subjects for each call type. **f** Same format as Fig. 3c, comparison of the activation start time between call types (Tukey boxplot). Data are from all recording sites showing significant activation (n = 10 sites for phee, n = 7 sites for trill, n = 23 sites for twitter, n = 6 sites for trillphee). Medians: horizontal lines inside the boxes. First and third quartiles: lower and upper borders of the boxes. Inner fences: whiskers outside of the boxes. Outliers are not plotted if any. An asterisk above a whisker indicates that the median is significantly lower than zero ($p = 0.014$, $0.047$, $7.3 \times 10^{-4}$ for phee, trill, and twitter, two-sided signed-rank test). There is no significant difference in the activation start time across the four types of calls ($\chi^2 = 4.6$, $p = 0.20$, Kruskal−Wallis test). **g** Same format as Fig. 2i, spatial distribution of the theta-band modulation of twitter calls for Subject M93A. The LFP power is calculated within an analysis window covering the duration of each twitter call. A black border on a circle indicates significant modulation. **h** Same format as **g**, for Subject M9606. R rostral, C caudal, D dorsal, V ventral. Source data are provided as a Source Data file.

within the call duration for each site, we found significant activation in two sites for phees, five sites for trills, fifteen sites for twitters and two sites for trillphees in Subject M93A, plus two sites for twitters in Subject M9606 (Fig. 4e). For the sites with theta-band activation before or during the vocalizations, the median activation start time (i.e., the earliest time at which theta-band LFP power showed significant activation, see Methods) is before the vocal onset for phee, trill and twitter calls (Fig. 4f, $p < 0.05$, two-sided signed-rank test), and close to the vocal onset for trillphees. There is no significant difference in median activation start time between the four call types (Fig. 4f, $p = 0.20$, Kruskal−Wallis test). For twitter calls, we found that the activation in the theta-band LFP power was more prominent towards the dorsal side of the arrays, near the primary motor cortex, in both subjects (Fig. 4g, h). For Subject M93A, activation was found across several cortical regions, with the strongest activation in the primary motor cortex (Fig. 4g). For Subject M9606, the activation was found in two sites at the border of the dorsal premotor cortex, Brodmann area 8 C and primary motor cortex (Fig. 4h).

It is noteworthy that a twitter call is composed of a series of discrete phrases each of which is an upward FM sweep (Figs. 2c and 5a). The repetition rate of the twitter phrases is ~7 Hz according to a large-

scale vocalization analysis study[34], which is within the frequency range of the theta-band. We wondered whether the LFP activity in the theta band is related to the repeated production of the twitter phrases. As shown by an example waveform from one cortical site (Fig. 5a), the theta-band LFP waveform showed oscillations at the same repetition rate as the twitter phrases. The onset of each twitter phrase (referred to as "syllable" in the text below) occurred at a particular phase of the LFP oscillation−near the trough of each cycle. Figure 5b (top) shows a few more example LFP waveforms and the corresponding twitter calls, revealing that this relationship is consistent across individual twitter calls. The alignment between syllable onset and the LFP phase held up even when the interval between two syllables became longer or shorter (e.g., first trace, Fig. 5b). Sometimes the syllable production skipped one cycle and the next syllable still occurred at a similar phase of the oscillation (Fig. 5b, second trace from top). These observations suggest phase-locking between twitter syllables and the theta-band LFP oscillation at this site. Figure 5b (bottom) shows the distribution of the theta-band LFP phase corresponding to the onset time of each twitter syllable for all twitter syllables at this recording site. We used the vector strength (VS) to quantify the degree of the phase-locking, i.e., how well each twitter syllable synchronizes with the theta-band LFP

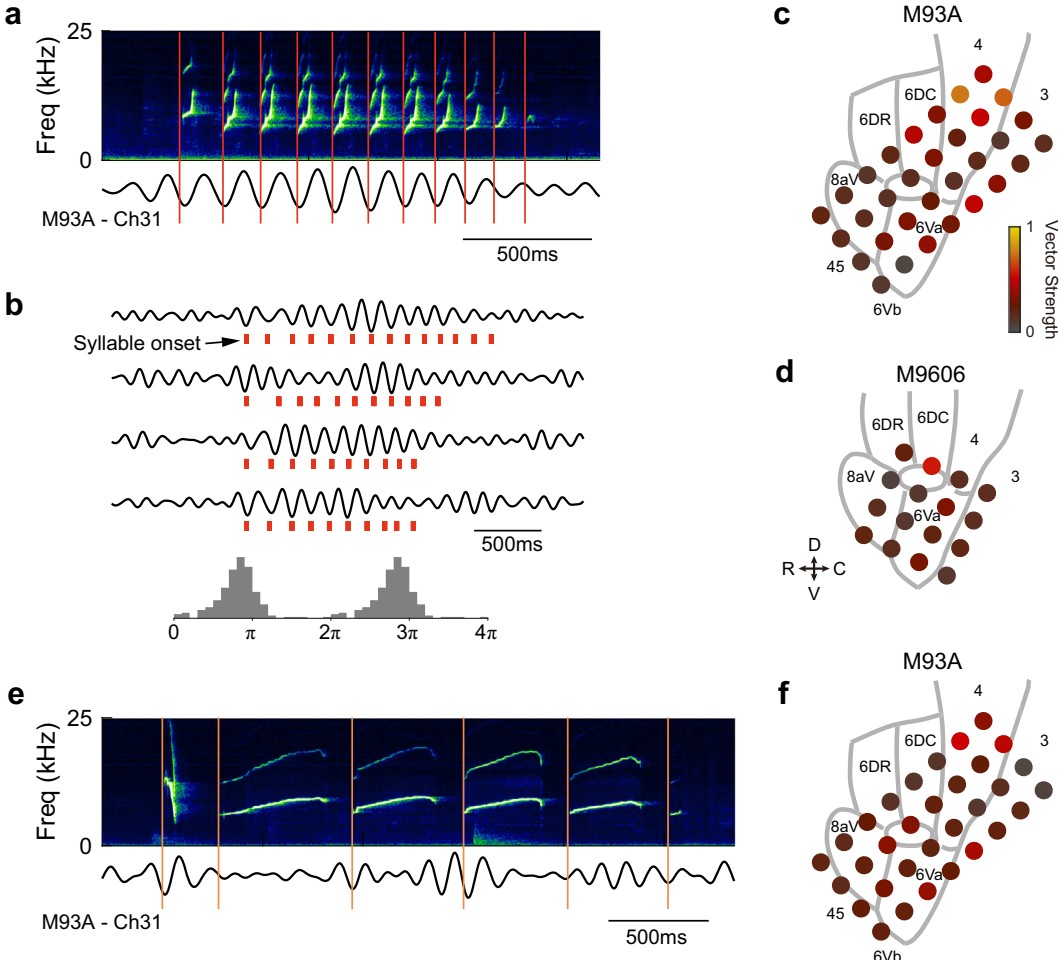

**Fig. 5 | Phase lock of the theta-band LFP oscillations to the individual phrases (also referred to as syllables) of calls. a** Top: spectrogram of one example twitter call. Bottom: theta-band LFP trace from one example recording site time-aligned to this twitter call. Vertical lines indicate the onset time of each syllable. **b** Top: four example traces showing the theta-band LFP oscillations from the same recording site as in **a**. Bottom: histogram of the phase of the LFP oscillations corresponding to the onset time of each twitter syllable from all twitter calls at this recording site. **c** Spatial distribution of the vector strength across all recording sites for twitter calls (Subject M93A). **d** Same format as **c**, for Subject M9606. R rostral, C caudal, D dorsal, V ventral. **e** Same format as **a**, for an example compound call. This call started with an sd-peep, followed by four phee syllables and an sa-peep. **f** Same format as **c**, vector strength for compound calls (Subject M93A). Source data are provided as a Source Data file.

oscillation. For this recording site, VS is 0.76 ($p < 0.001$, Rayleigh > 13.8). Across recording sites, we found a majority of sites showed significant phase-locking ($p < 0.001$, Rayleigh > 13.8), except for one site in Subject M93A and one site in Subject M9606 (both in the ventral part of the arrays). In general, sites towards the dorsal part of the arrays showed larger VS, indicating a stronger phase-locking between individual twitter syllables to the theta-band oscillations in LFP (Fig. 5c, d).

We then asked whether phase-locking occurred for other call types that also have multiple syllables. One subject (M93A) frequently produced compound calls[34] composed of several syllables with different structures. We visually examined the LFP waveform at the same site as in Fig. 5a along with the spectrogram of an example compound call (Fig. 5e). The LFP waveform showed a similar phase-lock to the syllable onset of the compound call, as was observed for the twitter calls. Although an individual syllable may have a longer duration than a twitter syllable and may span across multiple cycles of the LFP oscillation, the following syllable still started near the trough of the LFP oscillation. Across recording sites, the strongest phase-lock was found near the dorsal part of the array (Fig. 5f) for compound calls, consistent with the spatial distribution found for twitter calls. Together, these data suggest that LFP activity could reflect the temporal dynamics of

vocalizations, potentially correlated with the production of sub-components of a call.

## Single neuron activities modulated by vocal production

In addition to analyzing LFP signals, we investigated single neuron activity during vocal production in the social communication context and specifically, whether neuronal activity differentiated different call types with distinct acoustic structures. In contrast to the antiphonal calling paradigm, marmosets produced a relatively small number of phee calls in the colony environment when they engaged in vocal exchanges with other conspecifics in this social setting. The number of phee calls associated with each neuron was usually too small for calculating averaged firing rates in single neurons. Therefore, we only included trill, twitter and trillphee calls in the analysis for single neurons. A neuron is considered being tested for a call type (trill, twitter or trillphee) if there were at least 15 calls of that type recorded from the neuron. In total, 151 single neurons were tested for at least one call type in our experiments (96 neurons in Subject M9606; 55 neurons in Subject M93A).

Figure 6 illustrates four example neurons whose firing rates were modulated by the production of trill, twitter and trillphee calls. In each

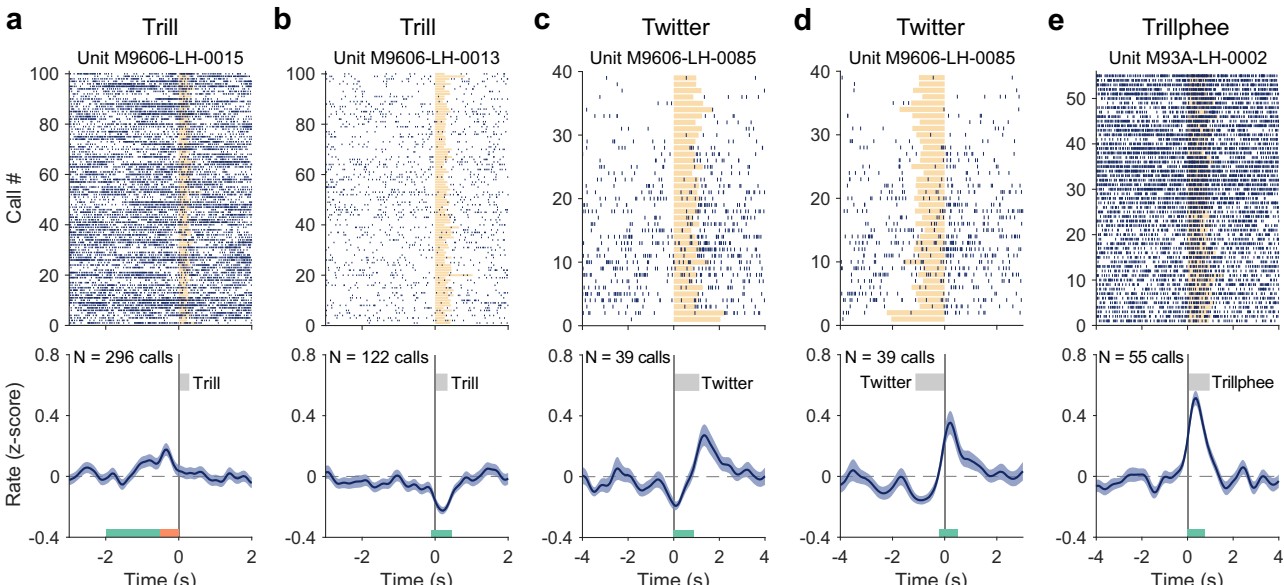

**Fig. 6 | Example neurons showing modulation in activities for trill, twitter and trillphee calls. a** Neuron showing increased activity before the vocal onset of trill calls. Top: Each vertical line indicates a spike. Spike timing is aligned to the vocal onset (time zero). Orange bars indicate the duration of trill calls. Only the first 100 calls are shown for clarity in the display (if the total number of calls is greater than 100). Bottom: Normalized firing rate (mean ± SEM). The firing rate is z-scored per trial and then averaged across trials (see Methods). *N* indicates the number of calls. The gray bar indicates the average duration of trill calls. **b**–**e** same format as **a**. **b** Neuron showing decreased activity during trill calls. **c** Neuron showing decreased activities near the vocal onset of twitter calls and increased activities after the end of twitter calls. **d** The same neuron as in **c**, aligned to the offset of twitter calls. **e** Neuron showing increased activities during the trillphee calls. The analysis windows are illustrated by colored bars at the bottom panels: early window (green in **a**), pre-call window (orange in **a**), during-call window (in **b**, **c**, **e**, showing the average location), post-call window (in **d**, showing the average location). Source data are provided as a Source Data file.

case, a relatively large number of calls was recorded. Neural firings were aligned to the vocal onset of each call in the analysis (Fig. 6, top panels). Because the recordings were made during free-roaming natural behavior in the colony room, we observed large variations in the firing rate across trials in many neurons (e.g., Figure 6c, top panel). To reduce the bias of firing variations across trials and reveal the modulation induced by vocalizations, we normalized the firing rate of each trial by z score and calculated the averaged normalized firing rate over all trials (Fig. 6, bottom panels, see Methods). The normalized firing rate profiles show modulations by vocal production in each of the four example neurons shown in Fig. 6.

To quantify these modulations, we calculated firing rates within four analysis windows (see Fig. 6 bottom panels; early: [−2, −0.5] sec, pre-call: [−0.5, 0] sec, during-call: [0 sec, 80% of call duration], post-call: [80% of call duration, 0.5 sec after call end], relative to the vocal onset) and compared these measures with that of the baseline window ([−8, −4] sec relative to the vocal onset) to determine whether and when a neuron was modulated by a particular call type (see Methods). The four example neurons shown in Fig. 6 illustrate the diversity of neural modulations when a marmoset vocalized. The neuron in Fig. 6a showed pre-vocal activation when the subject vocalized trill calls. In contrast, the neuron in Fig. 6b (trill calls) showed decreased firing rate after the vocal onset of trill calls. The neuron in Fig. 6c, d (twitter calls) showed both decreased firing rate (starting before the vocal onset and lasted till after the vocal onset, Fig. 6c) and increased firing rate in the post-call window (peaked right after the end of twitter calls, Fig. 6d). Finally, the neuron in Fig. 6e showed increased firing while the subject vocalized trillphee calls.

The total number of neurons modulated by each call type is summarized in Table 1. Overall, 31% (47/151) of neurons showed modulation by at least one of the three social communication call types. Figure 7a shows an analysis on when activation and suppression occurred relative to the four analysis windows for each call type. Interestingly, trill calls induced activations in the early window, before

the vocal onset and during vocalization (Fig. 7a, top). Twitter calls induced strongest suppression during vocalization and strongest activation after a call ended (Fig. 7a, middle). For trillphee calls, both activation and suppression occurred before the vocal onset, during and after the vocalization (Fig. 7a, bottom). As a control, we repeated the calculations with shuffled spike timing, which showed that the proportion of neurons in each analysis window is low (Fig. 7b). Therefore, the activation and suppression observed in the original data (Fig. 7a) did not arise from random fluctuations in spike activities.

**Distinct single-neuron activities across call types**

We further examined whether the activity of individual neurons in the marmoset frontal cortex could indicate different call types being vocalized. The largest number of calls produced by marmosets during social communication were trills. Twitters and trillphees were fewer. We, therefore, compared a neuron's activities between trill and twitter calls and between trill and trillphee calls, respectively. Trills and twitters have different acoustic structures. While trills are narrowband calls with sinusoidal frequency modulation (Fig. 2b), twitters are wideband calls with rapid upward frequency modulations and multiple syllables (Fig. 2c). If a neuron's activities showed similar modulations for trills and twitters, it is not likely to represent call type information. On the contrary, we found a subset of neurons showing different modulations between trills and twitters. The example neuron in Fig. 8a showed no modulation to trill calls but suppression during the production of twitter calls. Another example neuron showed activation for both trills and twitters (Fig. 8b). However, the activation for trills peaked before the vocal onset and the activation for twitters peaked near the end of calls.

While we studied 151 single neurons for at least one call type (at least 15 calls being vocalized) in our experiments, we were able to test only a subset of these neurons for more than one call type because the subjects may have produced an insufficient number of calls for one or two call types during the recording of a single neuron. A total

**Table 1 | Number of neurons tested and modulated for each call type**

| No. of of neurons | Trill | Twitter | Trillphee | Any |
|---|---|---|---|---|
| Tested | 151 | 65 | 60 | 151 |
| Modulated (Activated/Suppressed) | 28 (13/15) | 15 (6/10) | 16 (6/10) | 47 (22/26) |
| Proportion | 19% | 23% | 27% | 31% |

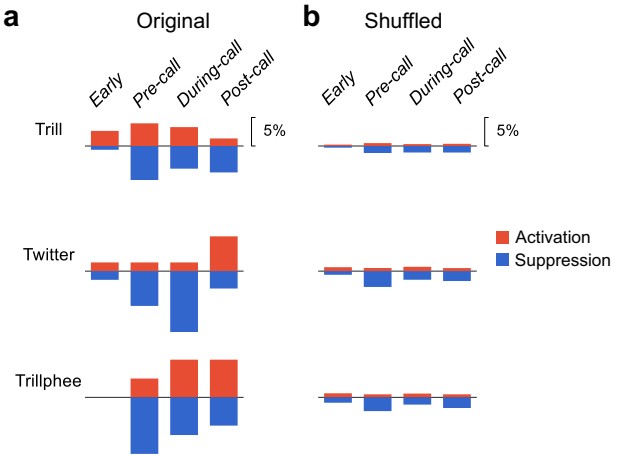

**Fig. 7 | Proportion of neurons showing modulation in different analysis windows. a** Comparison of the proportion of neurons showing activation and suppression between four analysis windows and three call types: trill (top), twitter (middle) and trillphee (bottom). The analysis windows are labeled on top. The proportion is calculated with respect to the total number of neurons tested for each call type (see Table 1). **b** The same comparison as **a**, with shuffled data (see Methods). Source data are provided as a Source Data file.

of 65 neurons were tested for both trills and twitters and 26 of them showed modulations by either call type (Fig. 8e). Among the 26 neurons, 11 neurons were modulated by trills but not twitters and 9 neurons were modulated by twitters but not trills. Six neurons were modulated by both call types, but two of them showed different modulations. Therefore, 85% (22 out of 26) of the modulated neurons showed a difference between the two call types. We then examined the location of these neurons and found that neurons modulated differently by trill and twitter were distributed across multiple cortical regions, rather than being within a confined spatial location (Fig. 8i, j, Supplementary Fig. 2a, b, e, f). The fact that a majority of neurons showed a difference in the vocal modulation between these two call types, with spatially distributed locations, suggested that the representation of call types in the marmoset frontal cortex existed in partially overlapping networks.

Next, we compared the modulation for trills and trillphees. These two call types are both narrowband calls, with the beginning part of trillphees resembling the structure of trills (Fig. 2d). One example neuron in Fig. 8c showed activation prior to the onset of trill calls (Fig. 8c, top). However, the same neuron showed no activation prior to trillphee calls but a trend of suppression near the call onset (Fig. 8c, bottom). Another example neuron showed no modulation to trill calls but activation towards the end of trillphee calls (Fig. 8d). In total, 60 neurons were tested for both call types and 25 neurons showed modulations by either type, 9 neurons by trills only and 11 neurons by trillphees only (Fig. 8g). Five neurons were modulated by both call types, but two of them showed different modulations. Therefore 88% (22 out of 25) of the modulated neurons showed a difference between the two call types. These numbers indicate that even between call types with similar acoustic structures, there are distinct neuronal

populations that represent each of them. Interestingly, nine neurons showed modulation by trills but not trillphees, even if trillphees shared a similar acoustic structure with trills in the beginning part. This suggests that call type may be represented as unique categories rather than by its spectrotemporal features in a subset of frontal cortex neurons in marmoset. The spatial location of neurons showing difference in modulation between trill and trillphee calls was also found to be distributed across cortical regions (Fig. 8k, l, Supplementary Fig. 2c, d, g, h).

Given the difference in vocal modulations at the individual neuron level, we postulate that the call type for each vocal production could be predicted from the activities of the neuronal population. We used a linear classifier with Monte Carlo simulations to decode call types between trills and twitters for each trial, using the activities drawn from all neurons that were tested with these two call types (see Methods). The classifier was trained separately by data in a sliding window (one second in length) near vocal onset and the performance was evaluated by classification accuracy using testing data in the same window (Fig. 8f). We found that when decoding call types between trills and twitters, the classification accuracy was significantly above chance level starting one second before the vocal onset and stayed above chance for most of the vocal duration till two seconds after the vocal onset (95% confidence interval of the mean does not overlap with chance level). Similar results were obtained when decoding call types between trills and trilphees (Fig. 8h). This analysis suggests that population activities within a short period of time around the vocal onset can distinguish the type of call about to be produced or being produced. It is interesting that the classification accuracy significantly increased before vocal onset, suggesting preparatory activities from the frontal network.

## Discussion

This study has provided insights on a long-standing question in the field of NHP neuroscience, i.e., the functional role of the monkey frontal cortex in social communication via vocal signals. Due to experimental challenges, it has been difficult to study neural activity of the frontal cortex while freely roaming monkeys are engaged in vocal communication with conspecifics using their full vocal repertoire. By applying wireless neural recording techniques, we were able to investigate neural correlates of the production of social communication calls in the frontal cortex of the marmoset. Based on single neuron activity and simultaneously recorded LFP signals from naturally vocalizing marmosets, we provided three pieces of key evidence to support a functional role of the frontal cortex in the production of social communication calls in marmoset monkeys. First, neural activities of particular regions of the frontal cortex exhibit differential signals for different types of marmoset social communication calls. This includes call-specific modulations in LFP beta-band power and spiking activities of individual neurons and population of neurons. Second, the activation and suppression of frontal cortex neuron occurred both prior to and during vocal production, suggesting potential roles of this brain region in vocal planning and execution. Third, theta-band LFP activities show temporal patterns that were phase-locked to temporal structure of twitter and compound call syllables. These findings expanded the previous studies in marmosets primarily based on phee calls and suggested an important function of the primate frontal cortex in the production of social communication signals across marmoset's vocal repertoire.

Oscillations in low-frequency brain signals like LFP have been known for a long time in both humans and non-human primates to be associated with motor movement. LFP beta-band power in the frontal cortex decreases when the subject starts to make voluntary hand movements or in arm reaching tasks (i.e., beta-band suppression), which occurred most prominently in sensorimotor cortex[39–41]. In our experiments, we observed widespread beta-band suppression when

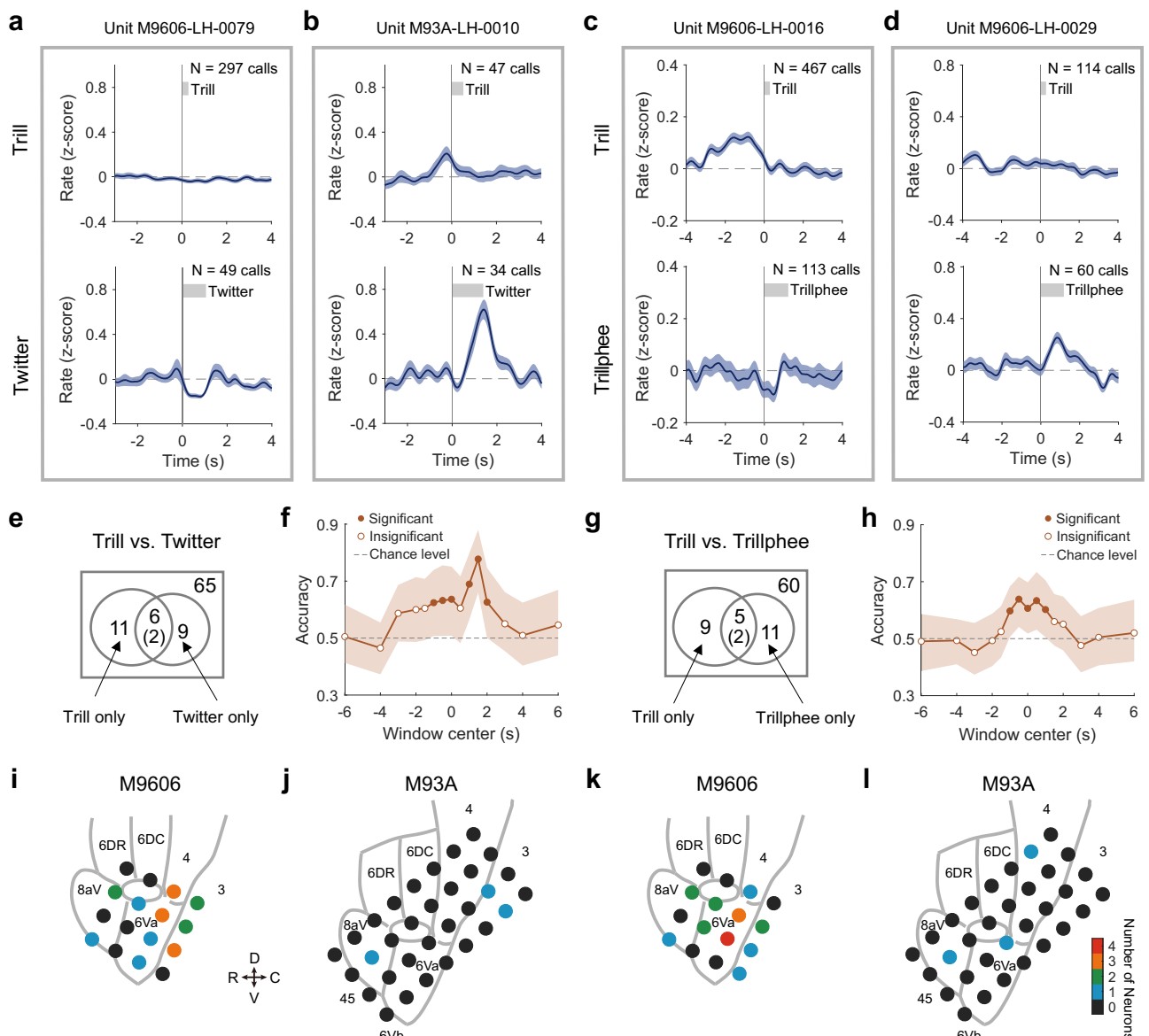

**Fig. 8 | Distinct activity for call types in single neurons and in the neuronal populations. a** Neuron showing different modulation between trill (top) and twitter (bottom) calls. The format of each panel is the same as Fig. 6a bottom panel (mean ± SEM). This example neuron showed no modulation for trill calls but decreased activities during twitter calls. **b–d** same format as **a**. **b** Neuron showing increased activities before the vocal onset of trill calls but near the end of twitter calls. **c** Neuron showing increased activities before the vocal onset of trill calls but not for trillphee calls. **d** Neuron showing no modulation for trill calls but increased activities during trillphee calls. **e** Venn diagram showing the relationship of neurons modulated by trills and neurons modulated by twitters. The number of neurons tested with both call types is indicated at the upper right corner of the rectangle. The circle on the left indicates the group of neurons modulated by trill calls. The circle on the right indicates the group of neurons modulated by twitter calls. The number in the overlapping region is for the neurons modulated by both call types. Within these neurons, the number showing differences between the two call types is indicated with parentheses. **f** Classification accuracy based on activities from neuronal populations to predict trill vs. twitter calls. Accuracy (mean and 95% confidence interval) is calculated using neural activities in a sliding window centered at different time points relative to vocal onset (time zero). **g** Same format as **e**, for neurons modulated by trills and neurons modulated by trillphees. **h** Same format as **f**, for predicting trill vs. trillphee calls. **i** Spatial distribution of neurons showing different modulation between trill and twitter calls for Subject M9606. Colors indicate the number of neurons found at each cortical site (color scale shown next to **l**). R rostral, C caudal, D dorsal, V ventral. **j** Same format as **i**, for M93A. **k, l** Spatial distribution of neurons showing different modulation between trill and trillphee calls. Source data are provided as a Source Data file.

marmosets vocalized four types of calls across the recorded regions in the frontal cortex. In general, the beta-band suppression started before the vocal onset (for phee, trill, and twitter calls) (Figs. 2 and 3c). Furthermore, the magnitude and time course of the beta-band suppression showed differences across the four types of calls (Fig. 3). Trill calls induced the smallest magnitude and shortest duration in the beta-band suppression, whereas phee calls induced the longest beta-band suppression. This matches closely to the vocal characteristics of these two call types in that trill is the softest and shortest call among the four

call types and phee is the loudest and has the longest duration[34]. Previous studies have found that LFP beta-band suppression showed correlations with sensorimotor processing[43], motor preparation and initiation[44–46]. Using grasping and reaching tasks, the pattern of beta-band modulation was found to be correlated with the specific types of movement[41,47–49], duration[50], context[51,52] and movement sequences[53]. Our data on marmoset vocalizations also showed correlations between the suppression and the motor output features, such as amplitude and duration, consistent with these previous findings. Interestingly, the

start of the beta-band suppression of phee calls occurred about half a second before vocal onset, much earlier than that of the other three call types (Fig. 3c). This suggests that the production of phee calls in social context may involve extended preparation, which may be needed as the complete vocal structure with long duration and multiple phrases is planned at call onset[54]. The early preparation may also result from a prolonged interval between the previous call from other individuals and the current phee call. Finally, given that beta-band suppression was found across many cortical areas, including prefrontal cortex, future studies should delineate whether beta-band activity in specific areas reflects vocal motor control, vocal initiation, or cognitive-level processing.

Rhythmic activities in the theta frequency band have drawn broad interests in human speech research. In most spoken languages, syllables occur with a repetition rate of 4–9 Hz[55], overlapping with the theta frequency range. Theta oscillations in the neural signals have been found in the human motor cortex and are thought to relate to sensorimotor processing in speech[56–58]. Theta activation is also observed when subjects are making compensatory adjustments when auditory feedback is suddenly altered[59]. In the motor control of hands, fine movements of fingers are found to be correlated with theta oscillations in the motor cortex[60]. A recent animal study has found theta-band signals in the rodent motor areas associated with skilled movements that require coordinated motor sequence during reaching and grasping[61]. In this rodent study, several key behavioral time points are found to be time-locked to the theta cycles[61]. In the present study, we observed theta-band activation in the frontal cortex of marmosets during vocal production of communication calls. An interesting observation was that the temporal fluctuations of the theta-band LFP waveform were phase-locked to individual syllables of twitter calls and compound calls. To the best of our knowledge, this is the first time such observations are reported in non-human primate vocal behaviors.

The twitter call of marmosets has a syllable repetition rate of 6–8 Hz[34]. Other less frequently produced marmoset call types with a repetitive structure have repetition rates between 4–10 Hz, such as peep-string phees[34] and fragmented phees[21,23,62]. Moreover, a recent study found that the mouth movement during marmoset vocal production followed the rhythm of the acoustically defined syllables and suggested that coordinated motor control for articulatory (e.g., mouth) and phonatory (larynx) systems gave rise to the rhythmic vocal output in marmosets[62]. Such coordination is found to be crucial for human speech production. The theta oscillation found here, which is tightly coupled to the instantaneous vocal output, provides evidence for a cortical neural correlate to the timing of marmoset vocalization components, such as individual phrases in a multi-phrase call or a compound call. It may also provide a neural basis for the coordination of articulators. Interestingly, a large portion of previous research on periodic movement patterns attributes the neural control to the "central pattern generator"[63] and it has been proposed that the vocal related counterpart is in the brainstem[64,65]. In contrast, we found vocalization-induced theta-band oscillation in the frontal cortex, a region likely to be associated with voluntary motor control, suggesting the possibility of a high-level control mechanism in marmoset vocal production. It remains an open question whether there are similar oscillation signals in the brainstem and whether the frontal cortex communicates with the brainstem to coordinate the oscillations.

Vocal communication through speech is perhaps the most important social behavior of humans. It has been a long-standing question of how the brain controls the generation of speech and other communication signals[66]. The general notion has been that the lateral part of the cortex, including frontal, parietal and temporal regions are involved in speech production and learning, whereas the medial structures in the cortex and the brainstem, including ACC and PAG, are involved in emotional vocalizations, such as cry and laughter[4]. Studies in non-human primates, however, have generated controversial results

in the past decades. Experiments using electrical stimulation and conditioned vocalizations suggested that the lateral frontal cortex of monkeys was dispensable for vocal production[11,67,68]. Experiments with single neuron recordings in macaques in operant conditioning paradigms found activities in the premotor and prefrontal cortex for conditioned vocalizations but not for spontaneous (i.e., self-initiated) vocalizations[13,14]. Recent studies in marmoset monkeys, on the other hand, have observed neural activities in the frontal cortex during self-initiated phee calls[30,31,33], suggesting the involvement of the lateral frontal cortex in vocal production[69]. The present study has expanded these earlier findings and provided further evidence to elucidate the function of the lateral frontal cortex in vocal production and social communication by marmosets.

Modulation of neural activities by different call types has not been studied in the frontal cortex before. Previous studies on the brainstem have found separate populations of neurons that are correlated with the production of several call types in squirrel monkeys[65,70,71]. In squirrel monkey PAG, a subset of neurons was found to be only activated to one or a subset of vocal types[71]. In ventrolateral pontine areas (VOC), which receive input from PAG and project to several cranial motor neuron pools involved in phonation, neurons were found to be only modulated by frequency-modulated (FM) calls but not non-FM calls[70]. In the downstream motor neuron pools, more than half of the neurons showed activities only to FM call types while another subset showed activities to both FM and non-FM calls[65]. These activities were thought to be related to the "central pattern generator"[72]. Further, anatomical tracing studies for PAG suggested that several upstream regions, including the hypothalamus and anterior cingulate cortex, may play a role in driving the different neuronal populations in PAG that were activated for different call types[73]. However, pharmacological inactivation of PAG was found to abolish vocal fold activity induced by the cingulate cortex but not by the laryngeal motor cortex[74], which provided evidence for two separate pathways of vocal fold control, one from the limbic cortex and the other from the neocortex. A recent study using retrograde tracing in marmoset laryngeal muscles revealed frontal cortical projections from the premotor and primary motor cortices[37]. In our experiment, we tested LFP and individual neurons' modulations for different call types in the frontal cortex. The fact that both LFP activity and individual neuron responses showed distinct patterns for different call types suggests that neural activities in marmoset front cortex are related to the generation of individual vocalizations. Therefore, our study provides support for a crucial function of the pathway involving the lateral frontal cortex, i.e., the category and feature of the vocalizations may be shaped by neural signals from the premotor or primary motor cortices, in addition to limbic or brainstem activities. Future studies may use electrical or optogenetic stimulation to test causal relationship between frontal activity and vocal production behaviors[75].

Interestingly, most neurons showing modulations in vocal production were modulated only by one call type. A subset of neurons was modulated only by trill calls but not by trillphee calls (Fig. 8g), despite the similarity between these two calls in the frequency modulation (trill part) in their call structure. A further hypothesis can be proposed that there exists an abstract representation of call type categories in a subset of neurons in the lateral frontal cortex. Future studies are needed to investigate how these neurons communicate with other neurons in the frontal cortex and in the brainstem nuclei.

## Methods
### Study design
Several components need to exist in order to study marmosets' natural vocal communication in a social context: (a) marmosets being recorded in a freely moving behavioral state to allow natural vocal production; (b) sufficient social contexts to elicit different types of social calls without conditioning or experimental reinforcement; (c)

techniques to record the neural activities and the vocal signals continuously and reliably from the marmoset without interfering with its behavior. These components require a systematic design in the experiment combined with a series of new techniques. Our lab has previously developed a behavioral paradigm to study natural vocal exchanges in a controlled experimental chamber[28], designed apparatus to allow wireless single-unit recordings in freely moving marmosets[38], and utilized tethered recording system in the marmoset colony to study the auditory cortex during vocal production and perception[76].

In this experiment, we developed new techniques based on the existing ones to achieve the goal of our study. We performed our recordings in the marmoset colony, which provided an enriched social context. During the period of this study, the colony room housed about 40–50 marmosets including breeding pairs and families with adults and babies. The recordings from the experimental subject were done either in its home cage or in a specially designed shielded booth. In either case, the subject was able to see other marmosets in the room and participate in vocal exchanges with them. While this environment provided the social contexts to elicit all types of vocalizations from the marmoset, it poses challenges to both acoustic and neural recordings. We will detail the solutions in the following paragraphs.

## Vocal communication behavior

Previous studies have shown that marmosets utilize more than ten types of calls in social communication both in the field[35,36] and in the lab setting[34]. Four of these types (phee, trill, twitter, and trillphee) are most frequently produced and have been well quantified in terms of their acoustic structures[34,77]. In the marmoset colony, individual animals housed there constantly exchange vocalizations in and outside of their home cages. The experimental subject, either tested in our recording apparatus (a shielded booth, as detailed below) or in its home cage, maintained vocal interactions with other individuals in the room. They produced the same range of call types as they are not being recorded and as in the census of general vocal repertoire in the marmoset population. The experimental subject either spontaneously generated these calls or in exchange with other individuals housed in the same room.

## Targeted acoustic recordings

Unlike a dedicated recording chamber, the marmoset colony does not have the sound isolation power ideal for acoustic recordings. Sound targets (individual marmosets) are relatively close to each other in spatial locations. Vocalizations produced span a wide dynamic range and are often overlapping in time. Figure 1f (middle) illustrates an example recording from the colony, where vocalizations from multiple marmosets and background noise are captured by the same microphone. The goal of the targeted acoustic recording is to overcome this challenge and obtain clean recordings of the vocalizations from the experimental subject (target).

The criteria for clean recordings should include the following. First, both loud and weak calls can be recorded. Since marmoset vocalizations have a relatively wide dynamic range, with some loud phees over 100 dB SPL and some trills below 50 dB SPL[78], the recording system needs to ensure that the weak calls will not be buried in background sounds and become un-detectable. The background sounds include two categories. One is room noise, from the air circulation system, marmosets' non-vocal activities in cages (e.g., jumping), and occasional human activities. The other is vocalizations from other marmosets. For example, a trill call from the target marmoset can easily be masked by a loud phee call from a neighboring marmoset. Second, the recording system should enable clear separation of vocalizations between target marmosets and non-target marmosets (source separation). For example, if a target marmoset is silent and a non-target neighbor makes a loud call, this call should not be mixed

into the target marmoset's vocalizations. Third, the recording system should facilitate precise and efficient segmentation during post-processing. Because of the large number of vocalizations generated and recorded in the colony, there is usually a heavy burden in the post-processing to segment calls and classify call types. It is favorable if the hardware design can reduce the load of post-processing. The key to fit these criteria is to increase the signal-to-noise ratio of the target marmoset vocalizations.

Traditional techniques in acoustic recordings have limitations in this scenario. For example, directional microphones may reject sound coming from the side but may not provide enough differentiation for sources in the front, especially when individual marmosets are housed in cages next to each other. One can presumably move the experimental subject to a separate cage far from its neighbors. However, this is less desired since the social context is changed as the distance increases between the experimental subject and other individuals in the room. Collar devices, small microphones mounted on an attachment to the animal, may obtain decent sound intensity from the experimental subject, but they require adaptation for the animal and more manual labor to attach them and to change the battery, causing interference to the recording sessions and potentially the subject's behavior.

Here we designed a parabolic-reference microphone pair to solve the recording problem (Fig. 1d). A parabolic microphone is composed of a parabolic reflector and a microphone placed at the focal point, which is often used in field recordings[79]. It selectively amplifies signals coming from the front and has a higher gain in high frequencies than in low frequencies. A reference microphone is the same microphone model used in a traditional way without a reflector, placed next to the parabolic microphone (both directed at the sound target). We used a parabolic reflector with a diameter of 20.5 inches, a depth of 6 inches and a focal length of 4 inches, made of transparent polycarbonate (generic supplier from eBay). A microphone (AKG C1000S, cardioid pick-up pattern) was pointed towards the reflector with its diaphragm placed at the focal point of the reflector. Theoretical calculations of the acoustic properties of the parabolic reflector[80] revealed that the lowest frequency cutoff for recording is about 650 Hz. Since the fundamental frequency of most marmoset vocalizations is above 5kHz[34], this parabolic microphone is well suited for the frequency range of marmoset vocalizations.

The pick-up pattern (the relationship between gain and angle) of the parabolic microphone is tested in a sound-attenuating chamber. The gain of the parabolic microphone was measured with pure tones of different frequencies (1 kHz, 2 kHz, 4 kHz, 8 kHz, 16 kHz) played from a speaker (KEF LS50) at a set of different angles (0°–180°; 0° incident angle means facing directly to the speaker) and was compared to the reference microphone (Fig. 1e, data for 8 kHz, interpolated by a spline function). The parabolic microphone (Fig. 1e, green curve) has a much sharper pick-up pattern than the reference microphone does (Fig. 1e, brown curve). For sounds coming from the front, the parabolic mic has nearly 20 dB additional gain than the reference microphone. As the sound source offsets in direction, the gain of the parabolic microphone quickly drops and then becomes smaller than the reference microphone. While a traditional microphone (used as the reference microphone) achieves about 10 dB front and back gain difference, the parabolic microphone achieves over 30 dB of such difference. This feature suggests two things. One is the parabolic microphone has superior directionality to a traditional microphone; the other is by comparing the intensity of signals recorded by the parabolic microphone and reference microphone, one can identify whether the source is located in the target direction.

In our recording setup, we placed this microphone pair in front of a target marmoset cage and calculated the difference of signal intensity between the two microphone channels. Given the dimension of the cage, the distance from the cage to the microphone and the pick-up

pattern, we obtained the threshold of intensity difference. If the signal intensity in the parabolic channel is higher than that in the reference channel by an amount larger than the threshold, the source is from the target marmoset cage. Figure 1f (top and middle) shows an example recording clip from the two channels. While both microphones pick-up vocalizations and noises in the room, signals from the reference microphone resemble a uniform background, whereas signals from the parabolic microphone enhance target vocalizations, as they are from the direction the microphones were faced at. Room noise and vocalizations from non-target locations have similar intensities in the two microphone channels, as they are mostly coming from the side. When subtracting the two spectrograms (Fig. 1f, bottom), room noise and vocalizations from non-target marmosets canceled out and only the vocalizations from the target marmoset were left (Fig. 1f, bottom, red circles). Using this setup, we can enable reliable detection of target vocalizations even when they are weak in intensity or overlapped by calls from other marmosets.

After recording, a custom-written Matlab program was used to segment target vocalizations from the continuous recordings based on machine learning algorithms. Call types were classified by the same program. All time points of vocalizations and call type labels were checked and corrected by human experimenters by visual inspection of the spectrograms.

## Wireless neural recording in the colony environment

Wireless neural recording techniques have previously been applied in the lab in a radio frequency (RF) shielded behavioral chamber[38]. We adapted this technique in the colony recording in a two-step process. In the first step, we used an analog wireless system (W16, Triangle Biosystems, or TBSI) in a custom-built RF-shielded booth to provide a supporting environment for reliable RF transmission (Fig. 1a). The booth is built with copper mesh and lined with RF absorption foam (EHP-5CV, ETS Lindgren) on most of the sidewalls and ceilings. The lower front part of the cage has no foam installed so the subject can see the colony through the booth. The experimental subject is placed in a plastic cage (60 cm × 41 cm × 30 cm) inside the booth made of plexiglass and nylon mesh. The wireless receiver is fixed directly above the plastic cage. The plastic cage is transparent to wireless signals and the shielded booth isolates the environment from interference in the colony room and reduces reflection of RF transmission from the walls of the booth. We tested this setup using the wireless headstage and a spectrum analyzer (R3172, Advantest). Within the space of recording, we obtained clean, reliable wireless transmission with stable signal strength, regardless of the headstage location or orientation, indicating there is no interference from outside of the booth or from multipath propagations inside the booth.

In the second step, we used a digital wireless system (W2100-HS32, Multichannel Systems, or MCS) to allow for recordings when the subject is free-roaming in its home cage (made with metal mesh and panels) (Fig. 1b). Digital transmission is known to be more robust to noise interference than analog transmission. Therefore, the recording location does not have to be protected by RF shielding. In our experiment, the receiver was placed above the subject's home cage. Before a recording session started, the orientation of the antennae was adjusted so that the real-time transmission quality was at an excellent level (above 95%, indicated in the Multi Channel Experimenter Software, MCS). This transmission quality was monitored throughout the recording sessions. In the rare case when there was data loss during transmission, the timestamps were logged by the recording software and the neural activities near that time period were excluded from the analysis. We used a custom-built battery pack (600 mAh, ~11 g) mounted on the headstage, which supported a continuous recording of 32 channels for up to 5 h.

For the analog system, raw neural signals were amplified (Lynx-8, Neuralynx), band-pass filtered (300–6000 Hz) and digitized at a 20 kHz sampling rate (PCI-6071E, National Instrument). Data were stored on a recording computer through a custom-written Matlab program. For the digital system, raw neural signals were band-pass filtered for spikes (300–6000 Hz) and the LFP (1–300 Hz) respectively and stored on a recording computer through a software of the wireless system (MC_Rack or Multi Channel Experimenter, MCS) at a sampling rate of 20 kHz. LFP signals were then downsampled to 1 kHz for analysis. Spike waveforms were sorted off-line with a template matching method in a custom-written Matlab program[30,38], with a minimum SNR of 15 dB. Neurons recorded from the same electrodes on the same day were considered the same units. Any artifact in spike or LFP signals was detected automatically in pre-processing and excluded from the analysis.

## Animal preparation and experimental procedure

Two adult marmosets were used in the experiment (M9606: 9 y.o., male, M93A: 2 y.o., male). M9606 was implanted with a 16-channel electrode array (Warp-16, Neuralynx) and M93A was implanted with a 32-channel electrode array (Warp-32, Neuralynx). Both arrays were in the left hemispheres. Implantation of the arrays followed a two-step procedure established in the lab previously. In the first step, marmosets underwent surgical procedures and were implanted with a head cap[81]. Marmosets were adapted to sit quietly in a custom-built restraint chair for a period of 2–4 weeks. During the surgery, the marmoset was initially anesthetized by an injection of ketamine (40 mg/kg) and acepromazine (0.75 mg/kg) and subsequently anesthetized with isoflurane (0.5–2.0%, mixed with pure oxygen). Before the skin incision on the head, 2 mL lidocaine hydrochloride (2%, vol/vol) was injected into the subcutaneous space. Under sterile conditions, two head posts were attached to the skull using dental acrylic. A thin layer of dental acrylic was applied to the skull covering part of the frontal-parietal cortex and part of the temporal cortex. The lateral sulcus was vaguely visible before the application of dental acrylic and was marked on the skull as a reference for anatomical locations. A thick layer of dental acrylic was applied to the rest of the skull to form a wall surrounding the frontal-parietal and temporal areas covered by the thin layer of dental acrylic. The thick layer of dental acrylic stabilized the head cap and later provided mechanical support for the electrode array. After the marmoset was fully recovered from the surgery, the electrode array was implanted as a separate second step. While the animal sat in the restraint chair, a craniotomy was created in the frontal areas by carefully drilling through the thin layer of dental acrylic and the underlying skull using a Dremel with a 0.5 mm drill bit. A second experimenter closely monitored the marmoset's condition, such as respiratory rate and movement. If any signs of discomfort were shown, ketamine (20 mg/kg) and acepromazine (0.75 mg/kg) were administered to sedate the animal briefly. The craniotomy was targeted at the premotor cortex, and the location was identified with reference to the marmoset brain atlas[82], using the lateral sulcus as a surface landmark. The craniotomy had a square shape for the 16-channel array and a rectangle shape for the 32-channel array. The array was positioned above the dura, sealed by Silastic (Qwik-Sil, WPI), and secured to the head cap by dental acrylic. A custom-built protective chamber was placed around the array and attached to the head cap with additional dental acrylic. The array was fully enclosed in the protective chamber and covered by a lid. After the implant procedure was finished, tungsten electrodes (4–12 MΩ, FHC or A-M Systems) housed in the arrays were advanced through the dura and adjusted in small steps (~50 μm) to search for single neurons. All experimental procedures were approved by the Johns Hopkins University Animal Care and Use Committee and were in compliance with the National Institutes of Health guidelines.

At the beginning of the experiment each day, the experimental subject was brought to a custom-built RF/EMI shielded chamber[83] and head-fixed in a primate chair. Neural signals were checked using the

wireless recording system. A polycarbonate protection cap was then mounted on the marmoset's head to protect the wireless headstage. The marmoset was then moved into the colony. The subject M9606 was recorded in a shielded booth located at the corner of the colony room. An analog wireless system was used for early sessions and a digital wireless system was used for later sessions (LFP signals were only obtained from these later sessions). A parabolic-reference microphone pair was placed 50 inches in front of the plastic cage. The subject M93A was recorded in its home cage. Before recording, the cage was moved to the side of the colony room next to the wall. Acoustic foams were placed in front of the wall near the cage to reduce sound reflections. The digital wireless system was always used for this subject. The parabolic-reference microphone pair was placed 40 inches in front of the home cage. To synchronize the neural and acoustic recordings, a pulse train with a period of two seconds was sent to the neural and acoustic recording devices simultaneously and was used to align the timing of the two sets of signals. A typical session lasted 2–5 h. After recording, the marmoset was moved back to the chamber. The wireless headstage was removed, electrodes were advanced[76] and the marmoset was sent back to the colony. In total, we recorded 21 sessions for subject M9606 and 31 sessions for subject M93A. We obtained stable recording for 1–8 neurons in each recording session.

## Data analysis

Pre-processing was first performed including segmentation of vocalizations in the acoustic recordings, classification of call types, spike sorting, and artifact detection. Calls overlapped with artifacts were excluded from subsequent analysis. To ensure temporal alignment of neural signals, any calls with a duration shorter than 200 ms were also excluded. This was done to prevent neural modulations from being washed out by averaging weak and transient activities for short trill calls together with more pronounced activities for longer calls.

The time-frequency representation of LFP power was calculated using the FieldTrip toolbox[84]. The spectrogram of LFP signals in each trial from a single electrode channel was first calculated by a wavelet transformation (Morlet wavelets with a width of 7) and then averaged across all trials (including all calls available for that channel)[85,86]. We chose a baseline window of [−3, −1] sec relative to the vocal onset to normalize LFP power. In the time-frequency representation, the power at each time point is normalized to the average power within this baseline window for each frequency band respectively. To calculate LFP power in a particular frequency band, the raw LFP signals were first band-pass filtered (12–30 Hz for beta band; 4–8 Hz for theta band) and then converted to analytic amplitude by a Hilbert transform. The signals were further downsampled to 200 Hz and the power was normalized with respect to the baseline window. To find the earliest time at which LFP power showed significant suppression or activation (suppression or activation start time), we used a sliding window of 100 ms long and compared the LFP power in this window to that in the baseline window. We selected the mid-point of the earliest window in which LFP power showed significant difference (two-sided signed-rank test) and was more than two standard deviations away from the power in the baseline window (see below for the details of the standard deviation calculation).

To characterize any modulation in LFP power in beta- or theta-band for a given electrode, an analysis window of [−0.1,0.2] sec relative to the vocal onset was used to quantify the LFP power. A recording site (or channel) was significantly modulated by a call type if the LFP power in the analysis window was significantly different from that in the baseline window (two-sided signed-rank test) and was more than two standard deviations away from the baseline LFP power. For the analysis of suppression or activation start time, we included three additional analysis windows: [−0.5,0] sec, call duration, and [0.3,0.9] sec. Sites showing significant modulation in any of these three windows or in the

[−0.1,0.2] sec window were included to ensure any early or late LFP modulations were captured. It is worth noting that using the standard deviation in the baseline LFP power as a criterion for modulation may induce bias, since the number of calls for each call type was largely different (e.g., there are a lot more trill calls than the three other call types), which would affect the size of standard deviation. To overcome this issue, we randomly drew 200 trials for each call type to calculate the averaged LFP power in the baseline window. We then used 200 ms non-overlapping windows (length comparable to that of the analysis window) to segment the averaged LFP power and calculated the standard deviation of the LFP power in these segments. This procedure was repeated 1000 times to bootstrap the mean standard deviation which was used as the criterion mentioned above.

To visualize the relationship of temporal profiles of beta-band suppression across call types and recording sites, we performed principal component analysis (PCA). The temporal profile of LFP power within the [−1.5, 2.5]s window was first downsampled to a 10 Hz sampling rate. PCA was then performed with temporal profiles from all call types and recording sites. The first three principal components explained 90% of the variance.

To quantify the phase lock of theta-band LFP to syllables of twitter calls or compound calls, we used band-pass filtered LFP waveforms (5–10 Hz, aiming to center around the twitter syllable repetition rate). We performed Hilbert transform to calculate the phase of the filtered LFP waveform at the vocal onset of syllables and then calculated the vector strength (VS)[87]. The higher the VS, the stronger the phase lock occurred. Rayleigh statistics was used to test whether the phase angle in the theta oscillation at which each syllable started has a non-uniform distribution. A compound call is a multi-syllable vocalization composed of a sequence of simple calls. Each individual syllable can be calls such as trills, phees or peeps.

For single-neuron analysis, we included neurons with at least 15 calls for subsequent analysis. To reduce the variation of baseline firing rate across trials, we z-scored the firing rate for each trial (relative to a 20-sec window centered at vocal onset). Firing rates were calculated in 50 ms time bins. To characterize the modulation of spike activities, we used four analysis windows (time relative to vocal onset): [−2, −0.5] sec (early activities); [−0.5, 0] sec (pre-call activities); [0 sec, 80% of call duration] (during-call activities); [80% of call duration, 0.5 sec after call end] (post-call activities). Firing rate in each of these windows was compared to that in a baseline window ([−8, −4] sec). A neuron is showing modulation in a specific analysis window if the firing rate in the analysis window is significantly different from that in the baseline window (two-sided signed-rank test) and is at least two standard deviations away from the baseline firing rate. Similar to LFP analysis, we used a bootstrap strategy to estimate the standard deviation of the baseline firing rate (using 500 ms segments). To get the number of modulated neurons with shuffled data, we circular-shifted the spike timing for each trial with a random amount and then calculated the mean firing rate across trials. The number of neurons with significant modulation with this shuffled data was calculated. This procedure was repeated 100 times to obtain the mean number of neurons with significant modulations (used in Fig. 7b).

For the population analysis, since the number of neurons recorded simultaneously during each session was small and the number of calls collected for each call type was different, we used a Monte Carlo simulation to balance the sample size. For a given neuron, twitter and trillphee calls may not have enough samples to compare with each other. Therefore, we built two types of classifiers to decode trill vs. twitter and trill vs. trillphee calls, respectively. For each type of classifiers, we included neurons with at least 20 calls to each call type and use the mean firing rate within a one-second long window centered at different time points relative to vocal onset to run the analysis (the classifiers are trained separately at each

window). We randomly select 70% of trials from each call type to form the distribution of training data and the remaining 30% to form the distribution of testing data. We then redraw these trials 5000 times for each call type to construct the actual training and test data set. We performed dimensionality reduction with PCA and projected data onto the top 10 principal components as features for the classifier. A linear discriminant analysis method was applied to classify the two call types. This procedure was repeated 500 times to obtain the mean and 95% confidence interval of the classification accuracy. Significance was determined if the confidence interval does not overlap with the chance level (0.5).

Two-sided signed-rank tests were used to test the significance of neural modulations. Kruskal–Wallis tests and post hoc analysis with the Bonferroni correction were used to compare the modulation size or time across multiple call types. Significance was determined at an α-level of 0.05.

### Histology
When all experiments were finished, electrolytic lesions were made by passing a small DC current through some of the recording electrodes (25 μA, 25 s). The animals were anesthetized by ketamine and euthanized by pentobarbital sodium. They were then perfused with a phosphate-buffered solution and 4% paraformaldehyde. Subject M9606's brain was sectioned in the coronal plane. A series of staining methods were applied to distinguish cortical regions, including Nissl, Myelin, and SMI-32 immunohistochemistry stains. To identify the locations of recording electrodes, the stained sections were scanned into digital images and reconstructed into a 3D model by comparing them to the standard marmoset brain atlas. The cortical regions were then segmented with reference to the standard atlas. The lesion marks were visually identified and assigned to one of the cortical regions in the segmented model.

### Reporting summary
Further information on research design is available in the Nature Portfolio Reporting Summary linked to this article.

## Data availability
All data needed to evaluate this study are provided within the paper and source data. The marmoset brain atlas used in the study can be accessed at https://www.marmosetbrain.org/. Raw data are available upon request to the corresponding authors. Source data are provided with this paper.

## Code availability
Custom MATLAB code used for data analysis is available through a public repository[88] at https://doi.org/10.5281/zenodo.8242611.

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

## Acknowledgements
We thank Reza Shadmehr and Cynthia Moss for helpful discussion on the data; Nathaniel Sotuyo, Shanequa Smith, Alexandra Prado, and Jessica Lynch for assistance with animal care; Kristina Nelsen, Marcello Rosa, and Brian Lee for help with histology and anatomical reconstruction; Calvin Qian, Emile-Victor Kuyl, Kevin Zhu, Hu Yi, for assistance with experiments and data preparation; Haowen Xu for optimizing the algorithm for call detection and classification. This work was supported by NIH Grant DC005808 (X.W.).

## Author contributions
L.Z. and X.W. designed the study. L.Z. developed the setup for targeted acoustic recording and wireless neural recording, performed experiments, and analyzed data. L.Z. and X.W. wrote the manuscript.

## Competing interests
The authors declare no competing interests.
