## [Peer Review File · Nature Communications]

REVIEWER COMMENTS

Reviewer #1 (Remarks to the Author):

Key results: In this study, the authors performed wireless electrophysiological recordings in prefrontal cortex of two freely moving marmoset monkeys while engaging in vocal communication with conspecifics in a social environment. They study neural activity underlying four different major call types of these animals to tackle the question if prefrontal cortical neurons in non-human primates play a role in encoding vocal signals. The authors find that local field potential (LFP) as well as single unit activity changes when the animals vocalize. More specifically, beta-band power seemed to be suppressed while theta-band power seemed to increase around call production. Another observation of this study is that theta-band power activity oscillates synchronously with twitter syllable production. Single unit activity shows modulation for all call types (except for phee calls) and includes pre onset, post onset activation and suppression. Strongest activations could be found in areas close to premotor cortex indicating that prefrontal cortex could play a role in vocal-motor signals. In summary, the current study is an important work in understanding neural correlates underlying social vocal behavior. The described methods as well as the provided analysis of the data seem to be well performed. However, there are certain points in which the authors could improve their experiment and analysis, and certain claims seem to lack the underlying data.

Validity: The authors suggest that call types may be represented as unique categories in the prefrontal cortex. A major concern that arises is that the observed modulated activity could be caused by something else. For example: Since the animals move freely around in their cage certain body movements (e.g. head movements) might be another source of this activity. Another example would be that these signals are driven by different arousal states of the animal. It is known, that distinct call types in marmosets are uttered in certain environmental and social contexts that could also drive distinct internal states of an animal. Prefrontal cortex has been shown to play an important role in arousal. A third example would be the following: Conspecific vocalizations can drive prefrontal activity in marmoset monkeys, and since conspecifics are present, vocalizations produced by them could also drive prefrontal activity. In my opinion, the authors do not rule out these possibilities in their study.

I think the provided data is not strong enough to support the suggestion that prefrontal cortex of marmosets is contributing “to the generation of the acoustic structure in the individual vocal signal”. An alternative explanation could be that the current arousal state of the animal is reflected in the neural data and causes a certain call type to be produced.

The authors claim, that the theta oscillation correlated to twitter calls “...is likely to underlie the control for the generation of syllables in marmoset calls...” However, the authors were only able to show this oscillation for only one call type. If this is true, an expansion of their analysis to other call types (e.g. phees, trills) should reveal more insight into this hypothesis. Furthermore, a large amount of previous studies found that the so called “vocal pattern generator” is located in the brain stem of

not just non-human primates but also many other mammalian species. The claim that the theta oscillation they found underlies vocal production is not supported by the data.

Significance: Considering the fact that speech and language are one of the most important forms of communication in humans, the understanding of cortical contributions to innate vocalizations of our closest relatives, non-human primates, is of significant interest for the understanding of how human speech and language evolved. Marmoset monkeys are a highly social and vocal primate species and an ideal model to study social vocal behavior.

Recording wirelessly from freely moving and behaving marmosets in a social context while interacting acoustically with conspecifics is important to study neural correlates underlying vocal behavior. Zhao and Wang performed experiments in marmoset monkeys which resemble a natural setting very closely.

The authors found theta-band LFP activity that seems to be phase-locked with the production of single twitter call syllables. These findings confirm previous behavioral studies suggesting that marmoset calls are built out of distinct short units that overlap with theta frequency range, similar to human speech rhythm. This is the first time a study provides physiological evidence that strengthens this hypothesis.

Data and methodology: The validity of the approach and the quality of the data are good, and the quality of presentation reads very well.

Analytical approach: The strength of the analytical approach, including the validity and comprehensiveness of any statistical tests is solid.

Suggested improvements:

- The authors should provide video data of the behavior of their animals during the experiments. This way the authors can rule out that prefrontal activity is caused by body movements or certain arousal states. Furthermore, it would be highly beneficial for the paper if the authors looked at prefrontal activity during vocalizations of conspecifics. These additional data analysis would help strengthen their arguments that prefrontal cortex plays a role in vocal production and they could confirm that neural activity is caused by self-produced vocalizations.
- Even though it is hard to perform more experiments when working with non-human primates, an additional experiment that could further strengthen the authors' arguments would be to compare neural activity between the production of a call and just listening to the playback of your own call. The experiment could be done by performing the same experiments and then playing back the animals own vocalizations afterwards. This way, the authors would be able to see if the neural modulation they observe is indeed a vocal-motor signal. An even stronger experiment to confirm the vocal-motor purpose of the observed signal would be to stimulate these brain areas and see if certain call types can be elicited or at least can be altered as it has been previously shown in the ACC of squirrel monkeys (Cramon and Jürgens, 1983).
- I would suggest that in addition to figure 3D the authors perform a more sophisticated analysis (e.g. PCA analysis and projection onto PC1 and PC2 in combination with the Euclidian distance

between data points). If different call types correlate with different beta-band powers, this might be a better way to preserve more information in the data. (Kingsbury et al. 2019).

- In line 201, the authors write “...theta-band activation was also observed for other call types, although with a smaller number of sites...” and don’t show examples of theta-band activation for phees and trill-phees in figure 4. It doesn’t seem that this is a reason to not show examples for these call types in figure 4, especially, since they showed already examples for beta-band activation in figure 2.
- In line 204, the authors write that theta-band power is smaller than for twitter calls, “...possibly due to much less intensity of trill calls...” Is this because louder calls have stronger theta-band activation than fainter calls? Please, provide an explanation why the authors think that.
- In figure 4E, it would be beneficial to provide a spectrogram of a twitter call type aligned with the LFP oscillation. Furthermore, the authors only look at twitter calls because of their syllable repetition rate of approximately 7Hz. It would be interesting to show if there is also underlying LFP oscillation with regard to the other call types as it has been previously suggested by reference 21. If this is the case it would further strengthen the authors’ argument of “a high-level control mechanism in marmoset vocal production”.
- In line 310, can the authors elaborate why these experiments were challenging?
- Why does figure 8A not have examples of trillphee calls?
- In lines 413-419: Do the authors suggest that the “vocal pattern generator” is located in the prefrontal cortex rather than the brain stem or do they think otherwise? Furthermore, in lines 417-419, what do the authors mean by “...whether there are similar oscillation signals in the brainstem from which the cortical oscillation is originated.”? Do they say that the brainstem drives cortical oscillations during vocalizations? Please, explain this in more detail!

Clarity and context: The text of the manuscript is clear and accessible and the authors provide sufficient context and consideration of previous work.

References: The manuscript references previous literature appropriately.

Reviewer #2 (Remarks to the Author):

The manuscript by Zhao and Wang describes the results of a study in which the authors investigated the role of frontal cortical areas in the production of different call types in common marmosets. The authors recorded single unit activity and local field potentials across a number of frontal cortical areas including 8aV, 45, 6Va, 6Vb, 6DC, 4, and 3 in two animals using microarrays and wireless recording techniques. Importantly, recordings were done in a more naturalistic context than

previous work, taking place in the marmoset colony, which encouraged the animals to make more frequent and diverse calls than have been investigated in previous work, which focused primarily on “phee” calls. The noisy colony context in which calls were recorded necessitated development of a novel methodology using parabolic and standard microphones to record, isolate, and identify calls produced by the subject animals. Neural activity surrounding “phee” calls, as well as “trill”, “twitter”, and “trillphee” calls was investigated to determine if differential signatures of these call types were present in marmoset frontal cortex. The authors observed call-specific changes in both LFP power and spiking activity across frontal areas. A differential suppression of power in the beta LFP band was found across many cortical areas, the timing of which additionally varied depending upon call type. Differential increases in theta-band power were also observed, particularly for “twitter” and “trill” calls, and theta oscillations were phase-locked to individual syllables of the “twitter” call. The authors additionally investigated the discharge rates of single frontal cortical neurons in time epochs surrounding the onsets of “trill”, “twitter”, and “trillphee” calls, as an insufficient number of “phee” calls were available for analysis. Call-type-specific activity was observed in this activity as well, with many neurons exhibiting increases or decreases in activity for a specific call type. The authors conclude that this experimental evidence supports a role of marmoset frontal cortex in vocal planning, initiation, and execution.

Overall, this work fills an obvious gap in the literature on social communication in the marmoset, using a novel vocal recording technique, coupled with wireless home cage recording to investigate the neural basis of the full vocal repertoire of freely moving marmosets. This has not been previously investigated due to the technical challenges inherent in recording and identifying calls made by subject marmosets in the home colony environment due to the constant production of calls from other marmosets and the difficulty inherent in unambiguously attributing specific calls to the subject animal. This laboratory has well-established expertise in these areas and the methodology and analytical techniques employed are sound. I outline some specific comments on the manuscript below:

1. Based on the magnitude and timing of LFP power changes and spiking activity, the authors posit a role of marmoset frontal cortex in both vocal initiation as well as production. In the case of initiation, I think that the authors should exercise some caution. Certainly, as the authors note, pre-movement decreases in beta power have been linked with movement initiation in other studies of motor control and the authors do a good job of linking their work to these findings. However, it must be noted that with respect to initiation of motor control, there are some specific criteria which must be met to validate this claim (Riehle & Requin, 1989, 1990, 1993). Not only must changes in activity precede the behavior in question, but the magnitude of activity must predict the probability of occurrence of the behavior as well as its timing. Obviously in this more naturalistic setting having the sort of control necessary to fulfill all of these criteria is essentially impossible. This does not invalidate the value of the work, but some care must be taken here, particularly considering the fact that some of the areas in which LFP power changes were observed were outside of those that might be considered to be linked directly to motor control and could instead be potentially more cognitive, for example areas 8 and 45. In subject M9606, some sites are also present in area 3. Given this, a general conclusion about initiation is problematic in my view.

2. The authors report a much earlier onset of suppression in beta power for “phee calls” than other call types. Is it possible that this is due not only to the content of the call but also a difference in the context in which it is made? Is it possible that the “phee” call is being made in response to a long duration call from another animal, such that it is simply planned further in advance than the other call types which could be self-generated more spontaneously, and that the modulation observed is driven by this context rather than the mechanics of the call itself?

3. On a related note, and shown in figure 3A, does the analysis window accurately capture the actual magnitude of maximal suppression of beta power for all call types? By eye it appears that capturing the full effect for twitter and trillphee as well as phee calls would reduce substantially the difference in magnitude here. This is important as it potentially affects the evidence supporting differences in beta power decreases for different call types.

4. A figure depicting the locations of the single units exhibiting differential activity for the different call types should be included. Was there any apparent organization of single units preferring different call types?

Typographical errors:

Page 3, line 7, “medical” should read “medial”. Similarly on line 54.

Page 4, line 70, “It is yet clear” should read “It is not yet clear”. Line 85, “exchanged” should read “exchanges”.

Page 5, line 87, “interferences” should read “interference”.

Page 12, line 247, “we also investigated”, could just read “we investigated”, since this was already qualified by saying “In addition”.

Page 14, line 302, “it’s not likely” should be reworded “it is not likely” to remove the contraction. Line 309 should read “more than one call type”.

Page 15, line 317 should read “Next, we compared...”

Page 16, line 350 should read “frontal cortex”. Same line “freely roaming monkeys are engaged...”
Line 351, “neural recording techniques”.

Page 17, line 370 “marmosets” should be plural.

Page 20, line 430 “in operant conditioning paradigms”.

Page 22, line 475 “interfering with”.

Page 29, line 648, “A typical session lasted 2-5 hours.” Same page, line 649 – should this read
“retracted” rather than “advanced.” ?

Page 32, line 715, “and the remaining 30%” is more correct.

Page 33, line 730, “They were then perfused” is more correct since there were two animals.

Stefan Everling

Reviewer #3 (Remarks to the Author):

The authors of this manuscript provide a crucial next step in the study of vocal production within NHPs. While prior studies have looked at trained and conditioned scenarios as in macaques,

marmoset research had advanced with "natural" communication. This communication was limited to the study of phee calls. Given the recordings happened in the colony, a whole new repertoire of calls have been recorded and analyzed. The natural production of these other calls are a necessary next step for marmoset vocal production research.

Other noteworthy results that bear repeating in no particular order:

1. The parabolic microphone set up is an ingenious way to isolate the subject's calls.
2. Using LFP, neural population level and single unit analysis, all showed there were differences in call production across the different call types (twitter, phee, trill, and trillphee).
3. The theta band locking to syllable production of twitter calls is unique and relevant to human syllable production.

I believe this work will be significant to the field of NHP vocal production and frontal cortex research. It's an important step forward given the call productions happened in naturalistic conditions.

Furthermore, the methodology is sound and in general straight forward, making it easy to replicate this research and expand upon it going forward.

There were some issues in the writing that I want to be addressed:

line 47, 54: Medical used but clearly meant medial.

line 498: any evidence to show that they make the same type of calls when recorded versus not? (note: I assume they would be would be helpful).

line 604: Mention of an "excellent level." Is that a proper setting? If not, it needs to be explained in more concise, replicable terms.

line 616: What units were kept and what was their dB and SNR quality for all the single units?

line 640: how many sessions were recorded in in the two monkeys? How many units were found per session?

Responses to Referees' Comments (NCOMMS-22-43584)

We thank the three reviewers for their constructive and helpful comments. In the revised manuscript, we have carefully addressed concerns raised by the reviewers (edits are made in "track change" mode). We hope the reviewers find our responses satisfactory and we'd be happy to address any further questions. The following is a point-by-point list of all changes made in response to the comments by the reviewers. The line numbers refer to those of the revised manuscript (please see the PDF file with track change markings for line numbers). The reviewers' comments are quoted first, followed by our response in blue font.

Reviewer #1 (Remarks to the Author):

Key results: In this study, the authors performed wireless electrophysiological recordings in prefrontal cortex of two freely moving marmoset monkeys while engaging in vocal communication with conspecifics in a social environment. They study neural activity underlying four different major call types of these animals to tackle the question if prefrontal cortical neurons in non-human primates play a role in encoding vocal signals. The authors find that local field potential (LFP) as well as single unit activity changes when the animals vocalize. More specifically, beta-band power seemed to be suppressed while theta-band power seemed to increase around call production. Another observation of this study is that theta-band power activity oscillates synchronously with twitter syllable production. Single unit activity shows modulation for all call types (except for phee calls) and includes pre onset, post onset activation and suppression. Strongest activations could be found in areas close to premotor cortex indicating that prefrontal cortex could play a role in vocal-motor signals. In summary, the current study is an important work in understanding neural correlates underlying social vocal behavior. The described methods as well as the provided analysis of the data seem to be well performed. However, there are certain points in which the authors could improve their experiment and analysis, and certain claims seem to lack the underlying data.

We thank the reviewer for recognizing the significance of this work! As the reviewer noted, the experiment in freely moving marmosets is important for understanding neural correlates underlying vocal behaviors in a social environment.

Validity: The authors suggest that call types may be represented as unique categories in the prefrontal cortex. A major concern that arises is that the observed modulated activity could be caused by something else. For example: Since the animals move freely around in their cage certain body movements (e.g. head movements) might be another source of this activity. Another example would be that these signals are driven by different arousal states of the animal. It is known, that distinct call types in marmosets are uttered in certain environmental and social contexts that could also drive distinct internal states of an animal. Prefrontal cortex has been shown to play an important role in arousal. A third example would be the following: Conspecific vocalizations can drive prefrontal activity in marmoset monkeys, and since conspecifics are present, vocalizations produced by them could also drive prefrontal activity. In my opinion, the authors do not rule out these possibilities in their study.

The reviewer raised several important questions here. As we all know, performing this type of experiments in freely moving marmosets in a social environment is essential because marmosets only vocalize with its full vocal repertoire in such conditions. At the same time, we do recognize, as does the reviewer, the complexity involved with such experimental conditions. We have performed additional data analyses and provided detailed responses to address the issues brought up by reviewer under the “Suggested improvements” section below, to address the issues brought up by the reviewer and provided alternative explanations of our results.

I think the provided data is not strong enough to support the suggestion that prefrontal cortex of marmosets is contributing “to the generation of the acoustic structure in the individual vocal signal”. An alternative explanation could be that the current arousal state of the animal is reflected in the neural data and causes a certain call type to be produced.

The additional analyses we have performed do not support the alternative explanation suggested by the reviewer above. Please see our detailed responses under the “Suggested improvements” section below. We have revised the above-mentioned sentence and the rest of that paragraph in the discussion to clarify what we meant (Line 593-600):

“The fact that both LFP activity and individual neuron responses showed distinct patterns for different call types suggests that neural activities in marmoset front cortex are related to the generation of individual vocalizations. Therefore, our study provides support for a crucial function of the pathway involving the lateral frontal cortex, i.e., the category and feature of the vocalizations may be shaped by neural signals from the premotor or primary motor cortices, in addition to limbic or brainstem activities. Future studies may use electrical or optogenetic stimulation to test causal relationship between frontal activity and vocal production behaviors.”

We’d like to point out that a recent anatomical study has shown descending projections from both the lateral premotor cortex and the medial SMA in marmoset cortex to brainstem vocal control structures (Cerkevich, Rathelot & Strick, PNAS 2022). This suggests that the premotor cortex is upstream to the brainstem vocal motor nuclei and may form parallel pathways together with SMA. We have now mentioned this evidence in the introduction and cited this work (Line 72-75):

“A recent anatomical study showed strong descending projections from the marmoset premotor cortex to downstream vocal control structures, suggesting potential functions of the premotor cortex in vocal motor skills”.

The authors claim, that the theta oscillation correlated to twitter calls “...is likely to underlie the control for the generation of syllables in marmoset calls...” However, the authors were only able to show this oscillation for only one call type. If this is true, an expansion of their analysis to other call types (e.g. pheeas, trills) should reveal more insight into this hypothesis. Furthermore, a large amount of previous studies found that the so called “vocal pattern generator” is located in the brain stem of not just non-human primates but also many other mammalian species. The claim that the theta oscillation they found underlies vocal production is not supported by the data.

Thanks for raising these important points! We have added analyses regarding the correlation between theta oscillation and other call types. We showed in the revised Fig.5 (E,F) that there

also exists phase-locking between syllable onset of multi-phrase compound phee calls and theta-band oscillations.

Regarding the “vocal pattern generator”, the reviewer seems to have over-interpreted our discussion in the previous manuscript. We did not suggest that the theta oscillation we observed represented a “vocal pattern generator” in the frontal cortex. We have revised the following passage in the discussion section (Line 539-543) as follows:

“The theta oscillation found here, which is tightly coupled to the instantaneous vocal output, provides evidence for a cortical neural correlate to the timing of marmoset vocalization components, such as individual phrases in a multi-phrase call or a compound call. It may also provide a neural basis for the coordination of articulators.”

Significance: Considering the fact that speech and language are one of the most important forms of communication in humans, the understanding of cortical contributions to innate vocalizations of our closest relatives, non-human primates, is of significant interest for the understanding of how human speech and language evolved. Marmoset monkeys are a highly social and vocal primate species and an ideal model to study social vocal behavior.

Recording wirelessly from freely moving and behaving marmosets in a social context while interacting acoustically with conspecifics is important to study neural correlates underlying vocal behavior. Zhao and Wang performed experiments in marmoset monkeys which resemble a natural setting very closely.

The authors found theta-band LFP activity that seems to be phase-locked with the production of single twitter call syllables. These findings confirm previous behavioral studies suggesting that marmoset calls are built out of distinct short units that overlap with theta frequency range, similar to human speech rhythm. This is the first time a study provides physiological evidence that strengthens this hypothesis.

Data and methodology: The validity of the approach and the quality of the data are good, and the quality of presentation reads very well.

Analytical approach: The strength of the analytical approach, including the validity and comprehensiveness of any statistical tests is solid.

We thank the reviewer for recognizing the significance of our work!

Suggested improvements:

- The authors should provide video data of the behavior of their animals during the experiments. This way the authors can rule out that prefrontal activity is caused by body movements or certain arousal states. Furthermore, it would be highly beneficial for the paper if the authors looked at prefrontal activity during vocalizations of conspecifics. These additional data analysis would help strengthen their arguments that prefrontal cortex plays a role in vocal production and they could confirm that neural activity is caused by self-produced vocalizations.

We appreciate the reviewer’s suggestions and have made efforts to perform additional analyses. We did not have video recordings for most of the recording sessions due to the complex setup we used for the wireless neural recording and acoustic recording with parabolic microphones in the colony room. In response to the reviewer’s inquiry, we are able to use a short video clip we recorded in the colony room from one subject (M9606) to test whether the vocal production is correlated with motor movements or overall arousal states (see Figures X1, X2, and X3 below).

Figure X1. This figure shows an example image from the video clip (left) and the extracted position and instantaneous speed of the animal’s head (right, showing a short segment). The duration of the video is about 33 min, with a frame rate of 15 fps, and resolution processed to 320x240 pixels. Scale bar at the upper left corner is 5cm approximately. The animal’s head position is tracked using the red color on the protection cap of the wireless headstage (yellow box on the left figure). Speed of the head movement is calculated using the x and y position at the center of the yellow box.

Within the time period where the animal’s head position can be successfully tracked (Fig.X1), 79 calls of the 4 major call types were recorded (phee: 3; trill: 57; twitter: 10; trillphee: 9, see Table X). Because the arousal state of the experimental subject cannot be directly measured in these experiments, we examined the animal’s movement states as a way to infer the animal’s arousal state. We used the average speed within a moving time window to quantify the trend in movements and identify potential states that spans over a relatively long timescale (e.g. dozens of seconds). We identified the time periods where the animal was either making active movements (“active state”) or was relatively still (“quiescent state”) (Fig.X2). Presumably the animal would be at a higher arousal level during active movements.

Figure X2. This figure shows the average speed in a moving window (60 sec long) within the entire video clip. We identified 3 periods with active state (orange color) and 2 periods with quiescent state (blue color). The dash line indicates the threshold (1 pixel/s). Gap within the curve indicates missing data where the animal moved outside of the camera view.

The total duration of active state is 7.3 min (22% of the total recording time) and the total duration of quiescent state is 26.0 min. The table below shows the number of calls and the call rate in each movement state. There were too few phee calls to calculate the call rate. The numbers of twitter and trillphee calls are limited so we focus further analysis on trill calls.

		Phee	Trill	Twitter	Trillphee	All Types
Number of calls	Active state	2	15	2	5	24
	Quiescent state	1	42	8	4	55
Call rate (calls/min)	Active state	-	2.06	0.27	0.69	3.02
	Quiescent state	-	1.62	0.31	0.15	2.08

Table X

The results we found here provide evidence to argue against the possibility that arousal states instead of call types determine the observed neural activity as we explained below.

(1) The subject produced all four call types in both active and quiescent states. Given that this video clip was recorded at a typical session which had the same configuration of animal cages in the colony as other recording sessions, it is reasonable to assume that the statistics found here is representative of all recording sessions in our study. The call rates during active and quiescent states are similar, for example, 2.06 vs. 1.62 calls/min for trill calls. A previous study found that the call rate of trills varied by several folds between different contexts and internal states (Combining Fig. 2B & Fig. S3 in Liao et al, PNAS 2018) which was not observed in the data shown on Table X. These statistics suggest that the production rate of calls in our recording sessions was not dominated by movement states or the related arousal states.

(2) We calculated the duration of trills in the two states and found that it was not significantly different (median: 0.25s in active state, 0.23s in quiescent states; $p = 0.97$, ranksum test). The dominant frequency of trills in the two states were also not significantly different (median: 6.8kHz in active, 6.6kHz in quiescent states; $p = 0.06$, ranksum test). A previous study found significant changes to duration and dominant frequency of trills due to context and internal states (Fig. 2D, 3D in Liao et al, PNAS 2018).

These data suggest that the overall movement level or arousal states within our recording contexts were not primary factors contributing to the vocal production and hence related neural activity in the frontal cortex of freely moving marmosets in this study.

We next analyzed the moment-to-moment change in behavior and movements by quantifying the instantaneous speed prior to and during each trill call. We ask whether there is any particular movement that is time-aligned to the call. We picked a fixed time period surrounding each trill call and plotted the instantaneous speed within this period (see Figure X3 below).

Figure X3. This figure illustrates the temporal dynamics of instantaneous speed around trill calls in quiescent state. The top panel shows the speed for each individual calls. Time zero is call onset. The blue curve/shading in the bottom panel is mean \pm sem of the actual speed aligned to call onset. The gray shaded region is the sem of randomly picked time segment from non-vocal periods (N=100) in quiescent state (as a baseline). Green bar: mean duration of trills.

We found that the instantaneous speed fluctuated around trill calls, with the peaks scattered over the time window. By comparing to the baseline temporal dynamics, we found that at no time point that speed around calls was significantly different from the non-call period ($p > 0.05$, ranksum test). Therefore, even if a neuron is modulated by the animal's movement, our analysis where trials are time-aligned to vocal onset is unlikely to be biased by the movement, as movement induced modulation would be washed out by the averaging. We were not able to repeat this analysis for other call types as we did not have enough number of calls in this video clip.

We appreciate the reviewer's question regarding neural responses to conspecific's vocalizations. We did not explicitly analyze it for this manuscript because we did not use targeted acoustic recording to specifically record vocalizations from other marmosets housed in the same colony room. One possible concern for the conspecific's vocalization is that it may overlap with the subject animal's self-produced vocalization. If neural activity was in response to calls from conspecifics, it may be mistakenly counted as modulated by vocal production. We have checked the spectrogram of our recordings for any possible overlaps. We found that when overlap occurred, the time period of overlap, call type and amplitude of conspecific calls were highly variable. Therefore, it is likely that any effects from overlapping conspecific calls would be washed out in analysis since we averaged neural activity across calls produced by the experimental subject.

An additional question one can ask is, whether neurons showing modulation by vocal production also show responses when listening to conspecific vocalizations. We believe that most of the neurons reported here do not show auditory response to conspecific's calls because our previous work showed that only a small proportion of neurons in the premotor cortex were responsive to conspecific phee calls (Roy et al, J.Neurosci, 2016).

Together, the evidence summarized above strengthens the interpretation that frontal activity is related to vocal production of specific types of marmoset calls, not due to other factors speculated by the reviewer.

- Even though it is hard to perform more experiments when working with non-human primates, an additional experiment that could further strengthen the authors' arguments would be to compare neural activity between the production of a call and just listening to the playback of your own call. The experiment could be done by performing the same experiments and then playing back the animals own vocalizations afterwards. This way, the authors would be able to see if the neural modulation they observe is indeed a vocal-motor signal. An even stronger experiment to confirm the vocal-motor purpose of the observed signal would be to stimulate these brain areas and see if certain call types can be elicited or at least can be altered as it has been previously shown in the ACC of squirrel monkeys (Cramon and Jürgens, 1983).

We thank the reviewer for suggesting these follow-up experiments! Although we did not present the playback of the animal's own call as an auditory stimulus, our data showed that a significant number of neurons exhibited response modulation prior to the vocal onset (Fig. 7A, early & pre-call windows), and the modulation to LFP in most cortical sites started prior to the vocal onset (Fig. 3C, 4F). The timing of the neural activity suggests that the activity is likely to be part of the vocal-motor program, rather than as a response to the sound which should occurred after the vocal onset.

As for electrical stimulation, it is beyond the scope of this manuscript. The observations reported in our study provide hints for future studies to use electrical or optogenetic stimulation methods to establish causal relationship between the frontal cortex and vocal production. We did not claim such a causal relationship in this manuscript. We have added the following sentence in the discussion to suggest future directions and we have cited the paper mentioned by the reviewer (Line 599-600):

"Future studies may use electrical or optogenetic stimulation to test causal relationship between frontal activity and vocal production behaviors (Cramon and Jürgens, 1983)".

- I would suggest that in addition to figure 3D the authors perform a more sophisticated analysis (e.g. PCA analysis and projection onto PC1 and PC2 in combination with the Euclidian distance between data points). If different call types correlate with different beta-band powers, this might be a better way to preserve more information in the data. (Kingsbury et al. 2019).

We have replaced Fig. 3D with new PCA results. This indeed provides a more convincing illustration of the distinction between call types. The original panel is moved to supplementary info.

- In line 201, the authors write "...theta-band activation was also observed for other call types, although with a smaller number of sites..." and don't show examples of theta-band activation for phees and trill-phees in figure 4. It doesn't seem that this is a reason to not show examples for these call types in figure 4, especially, since they showed already examples for beta-band activation in figure 2.

We have updated Fig. 4A-D with additional examples requested by the reviewer. In addition, we have re-organized Fig. 4 and 5 so that Fig. 4 focuses on theta-band activation, including the location in the array. Fig. 5 focuses on phase lock.

- In line 204, the authors write that theta-band power is smaller than for twitter calls, "...possibly due to much less intensity of trill calls..." Is this because louder calls have stronger theta-band activation than fainter calls? Please, provide an explanation why the authors think that.

We have now removed this postulation since we don't have sufficient data to prove it.

- In figure 4E, it would be beneficial to provide a spectrogram of a twitter call type aligned with the LFP oscillation. Furthermore, the authors only look at twitter calls because of their syllable repetition rate of approximately 7Hz. It would be interesting to show if there is also underlying LFP oscillation with regard to the other call types as it has been previously suggested by reference 21. If this is the case it would further strengthen the authors' argument of "a high-level control mechanism in marmoset vocal production".

We have added the spectrogram of a twitter call in Fig. 5A. One of the subjects (M93A) produced multi-phrase compound calls, in addition to twitter calls. There is indeed theta-band phase lock to syllables of these compound calls. We have now presented this finding in Fig. 5E, F. M9606 did not produce enough multi-phrase calls so we were not able to analyze phase lock.

We have described the phase lock with compound calls in the result section (Line 334-344) and revised the short discussion at the end of that paragraph (Line 343-344):

"Together, these data suggest that LFP activity could reflect the temporal dynamics of vocalizations, potentially correlated with the production of sub-components of a call."

- In line 310, can the authors elaborate why these experiments were challenging?

We mentioned that the experiments were challenging specifically for the purpose of comparing call types in single neurons. Marmosets produced trill calls most frequently and other call types not as frequently. Within the duration in which a single neuron can be isolated, it is likely that the animal did not produce enough vocalizations for calls such as twitter or trillphee so we were unable to compare the modulation to spikes between these call types and trill calls. We have changed this sentence to "we were able to test only a subset of these neurons for more than one call type because the subjects may have produced an insufficient number of calls for one or two call types during the recording of a single neuron." (Line 411-413)

- Why does figure 8A not have examples of trillphee calls?

We have now added examples of trillphee calls in Fig. 8C,D.

- In lines 413-419: Do the authors suggest that the "vocal pattern generator" is located in the prefrontal cortex rather than the brain stem or do they think otherwise? Furthermore, in lines 417-419, what do the authors mean by "...whether there are similar oscillation signals in the brainstem

from which the cortical oscillation is originated.”? Do they say that the brainstem drives cortical oscillations during vocalizations? Please, explain this in more detail!

We did NOT suggest that the “vocal pattern generator” is located in the frontal cortex instead of the brainstem. The discussion sentences in the previous manuscript might be unclear. We acknowledge that previous studies found evidence of “vocal pattern generator” in the brainstem. We believe our data provided new evidence for potential functions of frontal cortex in vocal production. It is possible that the brainstem has a theta-band oscillation and the oscillation in the frontal cortex is simply inherited from it. It is also possible that the frontal cortex generates signals that modulate brainstem activity while the brainstem activity still acts as the “vocal pattern generator”. We rephrased the sentences in the discussion section (Line 547-549) as follows:

“It remains an open question whether there are similar oscillation signals in the brainstem and whether the frontal cortex communicates with the brainstem to coordinate the oscillations.”

Clarity and context: The text of the manuscript is clear and accessible and the authors provide sufficient context and consideration of previous work.

References: The manuscript references previous literature appropriately.

Thank you!

Reviewer #2 (Remarks to the Author):

The manuscript by Zhao and Wang describes the results of a study in which the authors investigated the role of frontal cortical areas in the production of different call types in common marmosets. The authors recorded single unit activity and local field potentials across a number of frontal cortical areas including 8aV, 45, 6Va, 6Vb, 6DC, 4, and 3 in two animals using microarrays and wireless recording techniques. Importantly, recordings were done in a more naturalistic context than previous work, taking place in the marmoset colony, which encouraged the animals to make more frequent and diverse calls than have been investigated in previous work, which focused primarily on “phee” calls. The noisy colony context in which calls were recorded necessitated development of a novel methodology using parabolic and standard microphones to record, isolate, and identify calls produced by the subject animals. Neural activity surrounding “phee” calls, as well as “trill”, “twitter”, and “trillphee” calls was investigated to determine if differential signatures of these call types were present in marmoset frontal cortex. The authors observed call-specific changes in both LFP power and spiking activity across frontal areas. A differential suppression of power in the beta LFP band was found across many cortical areas, the timing of which additionally varied depending upon call type. Differential increases in theta-band power were also observed, particularly for “twitter” and “trill” calls, and theta oscillations were phase-locked to individual syllables of the “twitter” call. The authors additionally investigated the discharge rates of single frontal cortical neurons in time epochs surrounding the onsets of “trill”, “twitter”, and “trillphee” calls, as an insufficient number of “phee” calls were available for analysis. Call-type-specific activity was observed in this activity as well, with many neurons exhibiting increases or decreases in activity for a specific call type. The authors conclude

that this experimental evidence supports a role of marmoset frontal cortex in vocal planning, initiation, and execution.

Overall, this work fills an obvious gap in the literature on social communication in the marmoset, using a novel vocal recording technique, coupled with wireless home cage recording to investigate the neural basis of the full vocal repertoire of freely moving marmosets. This has not been previously investigated due to the technical challenges inherent in recording and identifying calls made by subject marmosets in the home colony environment due to the constant production of calls from other marmosets and the difficulty inherent in unambiguously attributing specific calls to the subject animal. This laboratory has well-established expertise in these areas and the methodology and analytical techniques employed are sound. I outline some specific comments on the manuscript below:

We thank the reviewer for recognizing the significance of this work!

1. Based on the magnitude and timing of LFP power changes and spiking activity, the authors posit a role of marmoset frontal cortex in both vocal initiation as well as production. In the case of initiation, I think that the authors should exercise some caution. Certainly, as the authors note, pre-movement decreases in beta power have been linked with movement initiation in other studies of motor control and the authors do a good job of linking their work to these findings. However, it must be noted that with respect to initiation of motor control, there are some specific criteria which must be met to validate this claim (Riehle & Requin, 1989, 1990, 1993). Not only must changes in activity precede the behavior in question, but the magnitude of activity must predict the probability of occurrence of the behavior as well as its timing. Obviously in this more naturalistic setting having the sort of control necessary to fulfill all of these criteria is essentially impossible. This does not invalidate the value of the work, but some care must be taken here, particularly considering the fact that some of the areas in which LFP power changes were observed were outside of those that might be considered to be linked directly to motor control and could instead be potentially more cognitive, for example areas 8 and 45. In subject M9606, some sites are also present in area 3. Given this, a general conclusion about initiation is problematic in my view.

Thanks for pointing this out and providing detailed explanation! We did not intend to claim that LFP power changes reflects initiation. We postulate that it is one of the possible functions. We now added the following sentences in the discussion to clarify our point (Line 507-510).

“Given that beta-band suppression was found across many cortical areas, including prefrontal cortex, future studies should delineate whether beta-band activity in specific areas reflects vocal-motor control, vocal initiation, or cognitive-level processing.”

We also revised the introduction and replaced “a call initiation signal” with “a high-level neural signal” (Line 79).

2. The authors report a much earlier onset of suppression in beta power for “phee calls” than other call types. Is it possible that this is due not only to the content of the call but also a difference in the context in which it is made? Is it possible that the “phee” call is being made in response to a long duration call from another animal, such that it is simply planned further in advance than the

other call types which could be self-generated more spontaneously, and that the modulation observed is driven by this context rather than the mechanics of the call itself ?

We agree it is possible that the early suppression may result from a long interval between previous call and current call. However, it is unclear whether the animal generated phee calls mostly in response to calls from other animals and it generated other call types mostly spontaneously. Since dozens of animals were housed in our recording environment, there could be more than one call from multiple individuals occurring before the call produced by the subject animal. Therefore, it is difficult to determine whether the subject's call was made in response to a call or which animal it was responding to. We added the following sentence in discussion to reflect this argument (Line 505-507).

"The early preparation may also result from a prolonged interval between the previous call from other individuals and the current phee call."

3. On a related note, and shown in figure 3A, does the analysis window accurately capture the actual magnitude of maximal suppression of beta power for all call types? By eye it appears that capturing the full effect for twitter and trillphee as well as phee calls would reduce substantially the difference in magnitude here. This is important as it potentially affects the evidence supporting differences in beta power decreases for different call types.

It is true the analysis window in Fig. 3A did not capture the maximal suppression for some call types. We chose this window because it captured the neural signal around vocal onset, which is hypothetically important for downstream structure to read out for vocal production. To strengthen the evidence, and following suggestions from Reviewer 1, we added a principal component analysis for the beta suppression (Fig. 3D,E). This analysis used information from the entire temporal profile of beta-band activity and clearly showed the distinction between call types.

4. A figure depicting the locations of the single units exhibiting differential activity for the different call types should be included. Was there any apparent organization of single units preferring different call types?

We have added this information in Fig. 8I-L and Supplementary Fig. 2. The spatial location of units preferring different call types appears to be distributed across brain regions.

Typographical errors:

Page 3, line 7, "medical" should read "medial". Similarly on line 54.

Corrected as suggested.

Page 4, line 70, "It is yet clear" should read "It is not yet clear". Line 85, "exchanged" should read "exchanges".

Corrected as suggested.

Page 5, line 87, "interferences" should read "interference".

Corrected as suggested.

Page 12, line 247, “we also investigated”, could just read “we investigated”, since this was already qualified by saying “In addition”.

Corrected as suggested.

Page 14, line 302, “it’s not likely” should be reworded “it is not likely” to remove the contraction. Line 309 should read “more than one call type”.

Corrected as suggested.

Page 15, line 317 should read “Next, we compared...”

Corrected as suggested.

Page 16, line 350 should read “frontal cortex”. Same line “freely roaming monkeys are engaged...”
Line 351, “neural recording techniques”.

Corrected as suggested.

Page 17, line 370 “marmosets” should be plural.

Corrected as suggested.

Page 20, line 430 “in operant conditioning paradigms”.

Corrected as suggested.

Page 22, line 475 “interfering with”.

Corrected as suggested.

Page 29, line 648, “A typical session lasted 2-5 hours.” Same page, line 649 – should this read “retracted” rather than “advanced.” ?

Corrected line 648 as suggested. For line 649, we used chronically implanted electrode array where individual electrodes can be moved deeper to search for new neurons. The detailed method is described in Eliades & Wang, J. Neu. Meth. (2008). We have now cited this paper after the word “advanced”.

Page 32, line 715, “and the remaining 30%” is more correct.

Corrected as suggested.

Page 33, line 730, “They were then perfused” is more correct since there were two animals.

Corrected as suggested.

Stefan Everling

Thank you for carefully reading our manuscript!

Reviewer #3 (Remarks to the Author):

The authors of this manuscript provide a crucial next step in the study of vocal production within NHPs. While prior studies have looked at trained and conditioned scenarios as in macaques, marmoset research had advanced with "natural" communication. This communication was limited to the study of phee calls. Given the recordings happened in the colony, a whole new repertoire of calls have been recorded and analyzed. The natural production of these other calls are a necessary next step for marmoset vocal production research.

Other noteworthy results that bear repeating in no particular order:

1. The parabolic microphone set up is an ingenious way to isolate the subject's calls.
2. Using LFP, neural population level and single unit analysis, all showed there were differences in call production across the different call types (twitter, phee, trill, and trillphee).
3. The theta band locking to syllable production of twitter calls is unique and relevant to human syllable production.

I believe this work will be significant to the field of NHP vocal production and frontal cortex research. It's an important step forward given the call productions happened in naturalistic conditions.

Furthermore, the methodology is sound and in general straight forward, making it easy to replicate this research and expand upon it going forward.

We thank the reviewer for recognizing the significance of this work!

There were some issues in the writing that I want to be addressed:

line 47, 54: Medical used but clearly meant medial.

Corrected as suggested.

line 498: any evidence to show that they make the same type of calls when recorded versus not? (note: I assume they would be would be helpful).

Comparing to a previous study (ref 34) which found phee, trill, twitter and trillphee as the four major call types, the subject animals produced the same call types when recorded. We expanded this sentence in introduction to reflect this point (Line 111-113)

"The experimental subject produced all types of vocalizations within the vocal repertoire of marmosets and the call types included in analyses were consistent between the experimental subject and the general marmoset population".

line 604: Mention of an "excellent level." Is that a proper setting? If not, it needs to be explained in more concise, replicable terms.

We added more technical details in the parentheses (Line 758) after the term "excellent level" ("above 95%").

line 616: What units were kept and what was their dB and SNR quality for all the single units?

We performed spike sorting using a template matching method (ref 30, 37). We kept units that had SNR above 15dB. The figure below shows the distribution of SNR for all the units we included in this manuscript. We have added this detail in the Method section (Line 772).

line 640: how many sessions were recorded in the two monkeys? How many units were found per session?

We recorded 21 sessions for M9606 and 31 sessions for M93A. Between 1 and 8 units were recorded in each session. We have added these details in the Method section (Line 805-806).

REVIEWERS' COMMENTS

Reviewer #1 (Remarks to the Author):

All my concerns have been addressed and the authors did an outstanding work in answering my questions. The findings and conclusions that frontal cortex activity could play a role in vocal production in marmoset monkeys are well supported.

No additional evidence is needed!

I think this work will be of significance to the field and supports as well as extends findings from established literature.

Thomas Pomberger

Reviewer #2 (Remarks to the Author):

The authors have addressed all my previous comments.

Responses to Referees' Comments (NCOMMS-22-43584A)

Reviewer #1 (Remarks to the Author):

All my concerns have been addressed and the authors did an outstanding work in answering my questions. The findings and conclusions that frontal cortex activity could play a role in vocal production in marmoset monkeys are well supported.
No additional evidence is needed!

I think this work will be of significance to the field and supports as well as extends findings from established literature.

Thomas Pomberger

We thank the reviewer for carefully reading our revised manuscript and the response letter! Suggestions and comments from the reviewer have helped strengthening the work.

Reviewer #2 (Remarks to the Author):

The authors have addressed all my previous comments.

We thank the reviewer for carefully reading our revised manuscript and the response letter!